# EBES: EASY BENCHMARKING FOR EVENT SEQUENCES

## ABSTRACT

Event sequences, characterized by irregular sampling intervals and a mix of categorical and numerical features, are common data structures in various real-world domains such as healthcare, finance, and user interaction logs. Despite advances in temporal data modeling techniques, there is no standardized benchmarks for evaluating their performance on event sequences. This complicates result comparison across different papers due to varying evaluation protocols, potentially misleading progress in this field. We introduce EBES, a comprehensive benchmarking tool with standardized evaluation scenarios and protocols, focusing on regression and classification problems with sequence-level targets. Our library [1] simplifies benchmarking, dataset addition, and method integration through a unified interface. It includes a novel synthetic dataset and provides preprocessed real-world datasets, including the largest publicly available banking dataset. Our results provide an in-depth analysis of datasets, identifying some as unsuitable for model comparison. We investigate the importance of modeling temporal and sequential components, as well as the robustness and scaling properties of the models. These findings highlight potential directions for future research. Our benchmark aim is to facilitate reproducible research, expediting progress and increasing real-world impacts.

## 1 INTRODUCTION

The world we live in is constantly changing (Laertius, 1925). We continuously collect and analyze data to understand and navigate this dynamic environment. This ongoing data collection helps capture the evolving nature of reality and can be captured in sequential datasets, which can be further analyzed or used for modeling.

Various types of sequential data are usually approached differently based on their characteristics. One prevalent form of sequential data is time series, regular measurements of some processes. The uniformity of these intervals enables researchers to apply a wide range of developed techniques (Eckner, 2012). Measurements of some processes that are taken or observed at non-uniform time intervals lead to irregularly sampled time series (ISTS). Fewer methods exist specifically for ISTS (Eckner, 2012), and modeling them brings new challenges (Li & Marlin, 2020). However, modeling them has a considerable importance since they naturally occur in many real-world areas: ecology (Clark & Bjørnstad, 2004), astronomy (Scargle, 1982), climate (Schulz & Stattegger, 1997), biology (Eckner, 2012), medicine (Goldberger et al., 2000; Johnson et al., 2016; Reyna et al., 2020), geology (Fang et al., 2023) and finance (Bazarova et al., 2024).

Another widely explored temporal data type is a *stream of discrete events*. Intervals between events are random, and modeling the distribution of inter-event intervals is an essential task with many applications. Temporal point process (TPP) model is commonly employed to model streams of discrete events (Du et al., 2016; Mei & Eisner, 2017; Omi et al., 2019; Jia & Benson, 2019; Shchur et al., 2019; Zhang et al., 2020; Zuo et al., 2020; Zhuzhel et al., 2023; Song et al., 2024).

In this work, we focus on another type of sequential data, **event sequences** (**EvS**), which are sequences of observations made at irregular times characterized by numerical and categorical features. **EvS** can be viewed as a generalization of both ISTS and streams of discrete events. Examples of various types of **EvS** are illustrated in Figure 1. Many modeling tasks naturally arise when dealing with sequential

---

[1]We attach an archive with the code. The code will be publicly available after the conference decision.

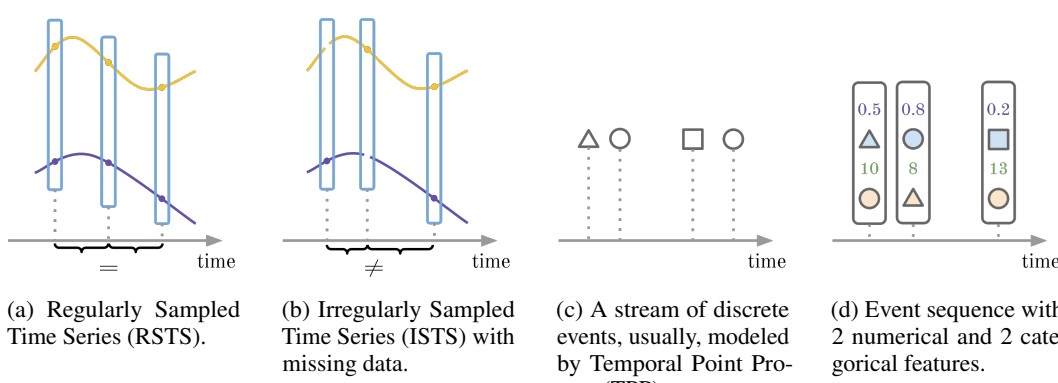

(a) Regularly Sampled Time Series (RSTS).

(b) Irregularly Sampled Time Series (ISTS) with missing data.

(c) A stream of discrete events, usually, modeled by Temporal Point Process (TPP).

(d) Event sequence with 2 numerical and 2 categorical features.

Figure 1: Sequential data taxonomy. Event sequences (**EvS**) generalize both irregularly sampled time series and streams of discrete events.

data, including whole sequence classification and regression (Shukla & Marlin, 2018), extrapolation or forecasting (De Brouwer et al., 2019), missing data imputation (Rubanova et al., 2019), point-wise classification (Hasani et al., 2022), and predicting the next event's time and type (Xue et al., 2024). Some of these tasks assume either a continuous or discrete nature of the data, which may not be known given a raw dataset. For instance, predicting the time of the next event is not reasonable when dealing with measurements from a continuous process. However, we can perform an assessment of the entire sequence regardless of the assumptions about the nature of the data.

As a task we consider the whole **EvS** classification and regression task, which we refer to *EvS assessment*. We emphasize the crucial role of **EvS** classification and regression in medicine (Shukla & Marlin, 2018), churn prediction (Jain et al., 2021), e-commerce (Zhao et al., 2023), fraud detection (Xie et al., 2022) and more.

**Our contributions are as follows:**

- We introduce EBES, a comprehensive benchmarking framework designed for **EvS** assessment. EBES features unified interfaces for datasets, models, and experimental protocols, facilitating future research in **EvS** assessment. Our library is publicly available.

- We design a benchmark protocol that considers both model and dataset analysis. Our evaluation includes various scenarios, including some specific to **EvS**, highlighting important properties of both the datasets and models.

- Using EBES, we evaluated various methods on established datasets through a multi-phase evaluation protocol. This approach ensures a fair and consistent comparison across different methods. All results are tested for statistical significance. As a result of our analysis, we provide recommendations for future research. These recommendations include possible pitfalls related to dataset usage and model evaluation.

## 2 BENCHMARK GOALS AND APPROACHES

Numerous methods have been proposed for **EvS** modeling and related problems. However, most of these methods lack rigorous evaluation, and there is currently no established benchmark for this domain. Benchmarking machine learning algorithms involves two main components: benchmark design and datasets, each presenting its challenges and goals. Below, we describe how we address each challenge in the context of **EvS**.

### 2.1 DATASETS

We have chosen three commonly used datasets based on previous studies Shukla & Marlin (2021); Udovichenko et al. (2024); Babaev et al. (2022); Moskvoretskii et al. (2024), one recent and one of the largest event sequence datasets MBD Dzhambulat et al. (2024), two medical datasets, and one synthetic pendulum dataset to validate the importance of time and how models capture the sequential properties of the data. We present statistics for each dataset in Table 1, and a detailed description of each dataset can be found in Appendix C.

Table 1: Statistics of sequential datasets used in our benchmark. The statistics are calculated on the train set if not specified otherwise. We use the following tasks notation: classification (C), regression (R) or multi-label classification (MLC). For MLC we report the average class balance.

| Dataset | Task | # classes | Class balance, % | Target | Category |
|---|---|---|---|---|---|
| PhysioNet 2012 | C | 2 | 86 / 14 | Mortality | Medical |
| MIMIC-III | C | 2 | 90 / 10 | Mortality | Medical |
| Pendulum | R | NA | NA | Air resistance | Physical (synth.) |
| AGE | C | 4 | 25 / 25 / 25 / 25 | Age group | Transactions |
| Retail | C | 4 | 27 / 21 / 27 / 24 | Age group | Transactions |
| MBD | MLC | 4 × 2 | 99.7 ± 0.2 / 0.3 ± 0.2 | Purchase items | Transactions |
| Taobao | C | 2 | 43 / 57 | Purchase event | E-commerce |

| | # seq. (train / test) | # events (train / test) | # events per seq. (mean ± std) | # cat. features | # num. features |
|---|---|---|---|---|---|
| PhysioNet 2012 | 4k / 4k | 299k / 299k | 75 ± 23 | 3 | 38 |
| MIMIC-III | 45k / 11k | 2.7m / 657k | 58 ± 93 | 1 | 10 |
| Pendulum | 80k / 20k | 2.5m / 631k | 32 ± 9 | 0 | 2 |
| AGE | 24k / 6k | 21m / 5.3m | 881 ± 125 | 1 | 1 |
| Retail | 319k / 80k | 37m / 9.1m | 114 ± 103 | 7 | 9 |
| MBD | 7.4m / 1.8m | 156m / 39m | 21 ± 435 | 11 | 1 |
| Taobao | 18k / 9k | 5.1m / 2.8m | 280 ± 387 | 2 | 0 |

**Data Quality**    One of the primary challenges in benchmarking is ensuring that the datasets used are high quality and accurately represent the problem domain. Poor data quality can lead to misleading benchmark results.

To address data quality, we employ two strategies:

- **Synthetic Dataset Development:** We create a synthetic Pendulum dataset, particularly useful for evaluating time-sensitive methods; dataset creation is described in Appendix C.
- **Dataset Analysis:** We analyze the correlation of model performance with Monte Carlo cross-validation. Specifically, we consider the relationship between metrics across various folds and the holdout test set.

**Diversity of Datasets.** Datasets with similar structures but different domains can vary greatly. For example, financial transactions differ significantly from medical records. Additionally, datasets can vary in complexity and difficulty. Our work includes a diverse range: two medical, three banking, one retail, and one synthetic dataset.

**Volume of Data.** Large datasets enable models to capture the complexity and nuances of real-world phenomena, leading to more accurate and reliable predictions. Moreover, different algorithms scale differently as the data grows. To address this challenge, we included datasets of various sizes.

**Open Access to Data.** It is crucial that data is available to researchers worldwide for reproducibility, collaboration, and innovation. While many event-sequence datasets exist, we focus on open-access ones and welcome contributions from other domains to enhance our collection. For example, astronomical observations (Carrasco-Davis et al., 2019) are event sequences but are not openly accessible.

## 2.2 BENCHMARK DESIGN

Creating effective benchmarks is a complex task, which involves designing tests that accurately reflect the capabilities of machine learning models across different scenarios:

**Model evaluation.** Hyperparameters are a fundamental aspect of machine learning that directly impacts model performance. However, the procedure of hyperparameters tuning is rarely described. Therefore, this becomes a source of non-reproducibility Arnold et al. (2023); Gundersen et al. (2022). Moreover, manual hyperparameter tuning can lead to the leakage of the test set into the training procedure and performance Lones (2021), and testing different hyperparameter values is necessary to find a model that generalizes well Gundersen et al. (2022).

In our procedure, we first conduct an extensive hyperparameter search. Randomness can destabilize models, causing large variances in results across training runs. Ignoring this sensitivity can create a false perception of research progress (Pecher et al., 2024). Therefore, after determining the optimal hyperparameters, we perform Monte Carlo cross-validation (MCCV) (Xu & Liang, 2001) with 20

seeds. At each MCCV step we train the model with the best hyperparameters and pick the checkpoint based on the (randomly sampled) validation set. Finally, the checkpoint is evaluated on the held-out test set. The mean test score across all seeds is reported as the model performance.

**Scalability**    As datasets grow larger, machine learning algorithms scale differently. Large datasets enable models to capture real-world nuances, improving prediction accuracy. To address this, we study the scaling properties of various event sequence assessment algorithms.

**Importance of Time and Sequence Order**    It is possible to perform **EvS** assessment while disregarding the temporal and sequential nature of the data. To evaluate the importance of each component, we designed two stress tests for event sequences: rearranging the sequence order and replacing time components with noise. This analysis provides significant insights and highlights future research directions.

**Model Granularity**    As AI systems grow more complex, assessing which components contribute to success becomes challenging. In our work, we evaluate different components, such as various aggregation approaches along the temporal dimension, batch normalization, and the impact of adding time as a separate feature on overall model performance.

### 2.3 BENCHMARK ACCESSIBILITY AND MAINTENANCE

The rapid evolution of machine learning makes keeping benchmarks up-to-date challenging. Benchmarks must reflect the latest advancements, incorporate new data and algorithms, and be maintained over time. Our work focuses on developing an easy-to-use plug-and-play codebase to facilitate collaboration and research. The library's interface structure enforces the independence of implementing new datasets, methods, and experiments, making adding and testing new components easy. We are committed to maintaining this benchmark and encourage contributions from researchers and practitioners to support reproducible research.

### 2.4 MODELS

We have carefully selected a diverse set of popular models and approaches that have been previously applied for **EvS** assessment. Some of the models, such as **MLP**, are included as baseline solutions, some are commonly used for sequential data **GRU** (Chung et al., 2014), **Mamba** (Gu & Dao, 2023), **Transformer** (Vaswani et al., 2017). The following models were explicitly designed to handle the unique challenges associated with **EvS**: **mTAND** (Shukla & Marlin, 2021), **PrimeNet** (Chowdhury et al., 2023) and **CoLES** (Babaev et al., 2022). Appendix B provides a detailed description of each model.

## 3 BENCHMARKING METHODS

### 3.1 DATASET PREPOSSESSING

In our work we aim to perform as little preprocessing as possible to preserve the originality of the data in order to prevent data preproccessing from affecting model evaluation. For ease of extensibility, we convert all datasets into a single format and release scripts that perform the conversion.

**Our data preprocessing includes:**

- Applying a logarithm to fat-tailed variables, which are selected manually;
- Rescaling time points to make the time range of all sequences to fall in $[0, 1]$;
- For missing values, we propagate them forward for the PhysioNet, MIMIC-III, and Pendulum datasets based on results in Che et al. (2018), and impute with constants for others.
- We encode categorical features using embedding layer and treat missing values as additional categories.

### 3.2 MODEL EVALUATION AND HPO

Hyperparameter optimization (HPO) and Monte Carlo cross-validation are at the core of our benchmark design, as they enable us to evaluate numerous design choices and hyperparameters, and to fairly compare models. Furthermore, we derive important insights from multiple HPO runs. Our evaluation procedure is twofold:

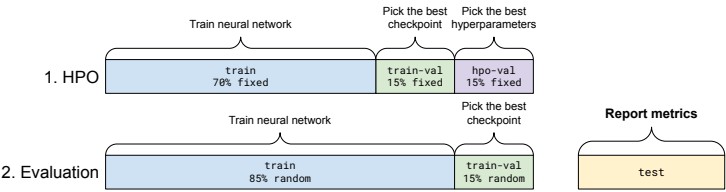

Figure 2: Data splits and their usage in our evaluation procedure.

- **HPO step,** here we perform hyperparameter optimization for all the models for each dataset. After obtaining the set of best hyperparameters (**BHP**), we use them for the next step.

- **Final evaluation,** during this step we train models with **BHP** 20 times using different seeds and random `train` and `train-val` splits. Final metrics are reported as average with standard deviation over 20 runs on test sets after models were trained from scratch.

A detailed algorithm with all the steps is outlined in Appendix D.

**Train-Val-Test splits**   For both steps we utilize data splits as follows: `train` - for training models, `train-val` - for early stopping procedure, we stop training if the model performance does not improve after several epochs and exceeds patience limit, `hpo-val` - a subset to evaluate the model to update HPO sampler, it does not present in **Final evaluation** step. Both `train-val` and `hpo-val` take 15% from the initial train dataset. See Figure 2 for clarification. For each split we apply on-target stratification. The number of patience steps is different for each dataset due to computational constrains.

For datasets, which do not have commonly accepted test sets, we cut 20% as our fixed test set. For HPO we use Optuna (Akiba et al., 2019) Tree-structured Parzen Estimator (TPE). For the main performance metrics of our benchmark, see Section 4.1.

## 4 EXPERIMENTS AND RESULTS

### 4.1 ASSESSMENT PERFORMANCE

In this section, we address the main question of the benchmark: **Which model performs the best?** The results are presented in Table 2, where methods are sorted from top to least performing. Along with the mean performance we report method's rank as a superscript. We performed pairwise Mann–Whitney $U$ test (Mann & Whitney, 1947) with Benjamini–Hochberg correction (Benjamini & Hochberg, 1995), methods with no significant performance difference ($p > 0.01$) share the same superscript. All top three performing methods are based on GRU with different pre-training strategies. CoLES improves metrics on tasks where the target is a characteristic of an observed sequence, such as Age, Pendulum, and Retail. However, on datasets where the target is somehow connected to future events, such as Taobao, MIMIC-III, MBD, and PhysioNet, the pretraining does not provide a significant boost. MLEM performs similarly to CoLES, likely due to its usage of pretrained CoLES components.

Transformer and Mamba comes next in rating, suggesting that this architectures are less suitable for **EvS** assessment. mTAND (Shukla & Marlin, 2021) excelled on the Pendulum dataset due to its architecture tailored for modeling the time component, particularly suited for ISTS like Pendulum. However, its poor performance on other datasets indicates that ISTS methods may not be as effective for general event sequences.

The MLP performs relatively well, typically within 5% of the top-performing method on all real-world datasets. This suggests that **EvS** assessment can be effectively carried out using aggregated statistics along temporal dimensions, a practice commonly employed in industrial applications with boosting models (ABIDAR et al., 2023). The difference in performance between MLP and mTAND on the Pendulum dataset further supports this idea, since we can not apply such aggregation approach to this dataset.

We see that, all methods show close results on the PhysioNet2012 dataset, based on ranks. This raises questions about its suitability for evaluating models for **EvS** assessment task.

Table 2: Model performance obtained using EBES. Results are averaged over 20 runs with the best hyperparameters determined through HPO. Statistically indistinguishable ($p > 0.01$) results share the same superscripts, indicating the method's rank for each dataset. The best-performing methods for each dataset are highlighted. Methods are sorted according to their average rank across all datasets.

| Dataset | Age | MBD | MIMIC-III | Pendulum | PhysioNet2012 | Retail | Taobao |
| Metric | Accuracy | Mean ROC AUC | ROC AUC | $R^2$ | ROC AUC | Accuracy | ROC AUC |
|---|---|---|---|---|---|---|---|
| **CoLES** | **0.634 ± 0.005**[1] | 0.826 ± 0.001[2] | **0.902 ± 0.001**[1] | 0.916 ± 0.004[2] | 0.840 ± 0.004[2] | **0.553 ± 0.002**[1] | **0.713 ± 0.002**[1] |
| **GRU** | 0.626 ± 0.004[2] | **0.827 ± 0.001**[1] | **0.901 ± 0.002**[1] | 0.896 ± 0.010[4] | **0.846 ± 0.004**[1] | 0.543 ± 0.002[2] | **0.713 ± 0.004**[1] |
| **MLEM** | **0.634 ± 0.003**[1] | 0.824 ± 0.001[3] | 0.899 ± 0.002[2] | 0.890 ± 0.007[4] | **0.846 ± 0.007**[1] | 0.544 ± 0.002[2] | **0.713 ± 0.004**[1] |
| **Transformer** | 0.621 ± 0.006[2] | 0.821 ± 0.002[4] | 0.894 ± 0.002[3] | 0.891 ± 0.015[4] | 0.838 ± 0.008[2,3] | 0.536 ± 0.006[3] | 0.692 ± 0.013[2,3] |
| **Mamba** | 0.609 ± 0.006[3] | 0.820 ± 0.003[4] | 0.895 ± 0.002[3] | 0.908 ± 0.005[3] | 0.835 ± 0.006[3,4] | 0.538 ± 0.003[3] | 0.693 ± 0.023[2] |
| **mTAND** | 0.582 ± 0.009[4] | 0.798 ± 0.002[6] | 0.888 ± 0.003[4] | **0.941 ± 0.009**[1] | 0.841 ± 0.005[2] | 0.519 ± 0.003[5] | 0.672 ± 0.010[4] |
| **PrimeNet** | 0.583 ± 0.011[4] | 0.780 ± 0.006[7] | 0.887 ± 0.004[4] | 0.842 ± 0.017[5] | 0.839 ± 0.004[2,3,4] | 0.521 ± 0.003[5] | 0.681 ± 0.010[3] |
| **MLP** | 0.581 ± 0.007[4] | 0.809 ± 0.001[5] | 0.881 ± 0.001[5] | 0.165 ± 0.005[6] | 0.835 ± 0.004[4] | 0.526 ± 0.002[4] | 0.659 ± 0.035[4] |

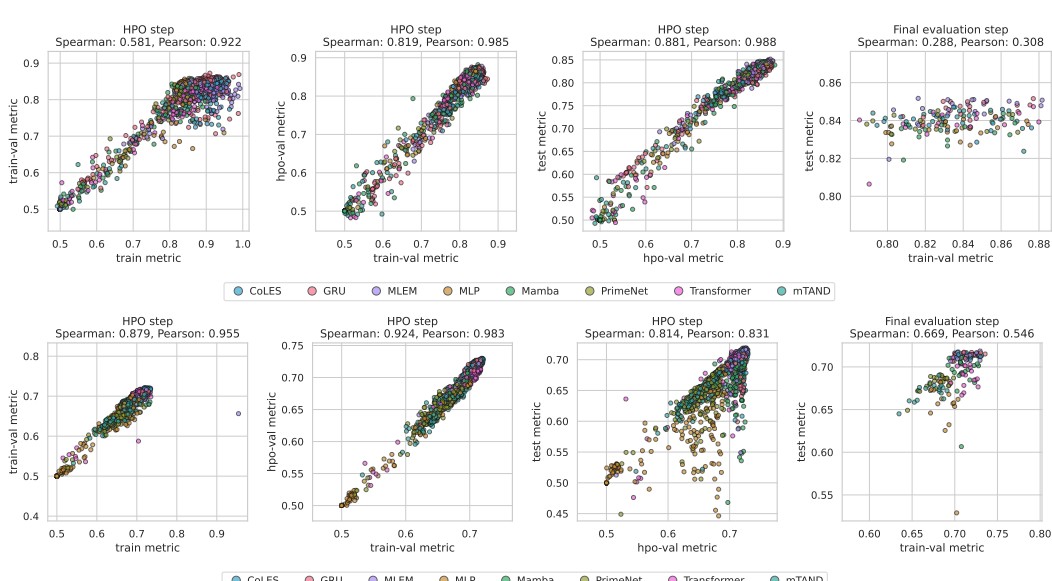

Figure 3: Performance metric relationships and correlations of different subsets among all methods on PhysioNet2012 (top row) and Taobao (bottom row) are presented. We do not observe a correlation between the test metric and `train-val` on PhysioNet2012, as seen in the right upper corner. For the Taobao dataset, we do not observe a clear linear trend between `hpo-val` and the `test` metric suggesting the presence of distribution shift.

## 4.2 DATASET ANALYSIS

In this section, we analyze datasets based on data from the **HPO step** and **Final evaluation** phases, exploring relationships between metrics from different data subsets. Correlations between different subsets for the PhysioNet2012 and Taobao datasets are depicted in Figure 3, with other datasets presented in Appendix E.

During the HPO step, we observe overfitting for most datasets, as `train` metrics increase while `train-val` metrics plateau, as seen in Figure 3 on the left. This supports the use of early stopping.

Metrics of `hpo-val` and `test` subsets (third column in Figure 3) are strongly correlated unless the test set is sampled out-of-time, as seen for the Taobao dataset. Here, `hpo-val` and `test` metrics lack a clear linear trend, but `train-val` and `hpo-val` metrics do, suggesting a distribution shift in the test set.

For most datasets, in the **Final evaluation** phase (fourth column in Figure 3), validation and test set metrics exhibit a linear trend, except for PhysioNet2012, where different validation metrics attribute to similar test metrics. This supports our observations in Section 4.1, where results for most models are not statistically distinguishable for most methods on PhysioNet2012.

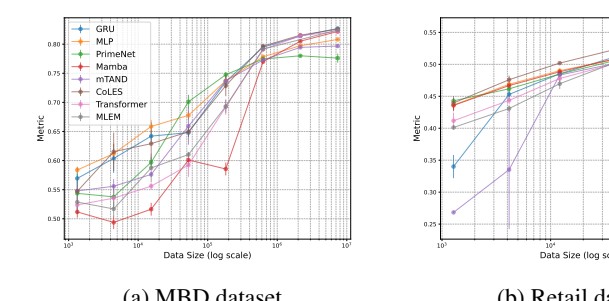

|  (a) MBD dataset  |  (b) Retail dataset  |

Figure 4: Performance of various models as a function of number of sequences. Metrics from Table 1 are reported. Number of sequences is presented in log scale. Standard deviation across 3 runs is depicted as vertical lines.

### 4.3 DATA SCALING RESULTS

To study the scaling properties of various models, we evaluated each model trained with different numbers of sequences. We focused on two biggest real-world dataset in our benchmark: Retail and MBD. We sampled different subsets, each containing progressively more data. Each model was trained from scratch on different-sized subsets with Monte Carlo cross-validation using three random seeds.

A common approach is to estimate model performance with a fixed data size. However, as seen in Figure 4, while all models improve with the growth of the data, their ranking does not stay the same, except for CoLES on the Retail Dataset, where it demonstrates superior performance. With some data size, even MLP becomes a top performer. Most models, except for MLP, mTAND, and PrimeNet, converge to similar performance on the MBD dataset given a large data size. It is worth noting that for each dataset, we used the **BHP** found for each model when the dataset was at its full size.

The standard deviation, decreases as the data size increases and models perform very differently with smaller subsets. This makes evaluating model performance on relatively small datasets more prone to misleading results.

### 4.4 ASSESSING ARCHITECTURE DESIGN CHOICES

Although our models exhibit a diverse range of architectures, there are several common design choices among them. We evaluated the impact of these choices as part of our HPO procedure.

We observe that some design choices depend more on the dataset than on the method, highlighting the importance of **HPO** for fair evaluation. First, we study the effect of different aggregation approaches along the temporal dimension on overall performance. We focus on two approaches: mean across all hidden states and the last sequence state. The best aggregation strategy depends more on the dataset than on the method. Similarly, batch normalization for numerical features improves performance for almost all methods and datasets, except for Pendulum. Finally, we evaluate the importance of hyperparameters according to **HPO**. There is no clear winner except for the learning rate, which is often the most important hyperparameter across all HPO runs. Results are presented in Tables 10, 9 and 8 in the Appendix.

### 4.5 IMPORTANCE OF SEQUENCE ORDER

One aspect of **EvS** is the order of events in a sequence. To examine its importance, we conducted two experiments: 1) We took models trained on regular data and evaluated them on test sequences with permuted order, keeping the time component unchanged. 2) We removed the time component and retrained the models on sequences with permuted order, then evaluated them on permuted test sequences.

**Testing on Permuted Sequences** We evaluated pre-trained models from the **Final evaluation step** on perturbed sequences. Missing values were filled prior to shuffling, and time was added as a

Table 3: Robustness to sequence permutation results. We report performance difference relative to metrics obtained on not permuted sequences. Models were train on non-permuted data; only the test set was permuted. Values with statistically significant difference ($p < 0.01$) in performance are highlighted and marked with asterisk.

| Dataset | Age | MBD | MIMIC-III | Pendulum | PhysioNet2012 | Retail | Taobao |
| --- | --- | --- | --- | --- | --- | --- | --- |
| Metric | Accuracy | Mean ROC AUC | ROC AUC | $R^2$ | ROC AUC | Accuracy | ROC AUC |
| **CoLES** | $-1.63\%^*$ | $-0.09\%$ | $-1.86\%^*$ | $-219.60\%^*$ | $-2.36\%^*$ | $-1.57\%^*$ | $-0.49\%^*$ |
| **GRU** | $-1.15\%^*$ | $-0.10\%$ | $-4.24\%^*$ | $-227.58\%^*$ | $-1.49\%^*$ | $-2.25\%^*$ | $-0.67\%^*$ |
| **MLEM** | $-1.52\%^*$ | $-0.30\%$ | $-1.43\%^*$ | $-242.09\%^*$ | $-1.71\%^*$ | $-2.57\%^*$ | $-0.89\%^*$ |
| **MLP** | $-0.00\%$ | $-0.00\%$ | $-0.00\%$ | $-0.00\%$ | $-0.00\%$ | $-0.00\%$ | $-0.00\%$ |
| **Mamba** | $-1.20\%$ | $-0.06\%$ | $-3.04\%^*$ | $-351.14\%^*$ | $-0.65\%$ | $-2.44\%^*$ | $-0.00\%$ |
| **PrimeNet** | $-7.82\%^*$ | $-4.08\%^*$ | $-3.72\%^*$ | $-128.39\%^*$ | $-3.95\%^*$ | $-26.41\%^*$ | $-2.12\%^*$ |
| **Transformer** | $-0.00\%$ | $0.00\%$ | $-0.00\%$ | $-5.20\%^*$ | $0.03\%$ | $-0.09\%$ | $-0.05\%$ |
| **mTAND** | $-8.95\%^*$ | $-5.05\%^*$ | $-5.05\%^*$ | $-133.66\%^*$ | $-4.13\%^*$ | $-28.09\%^*$ | $-4.13\%^*$ |

Table 4: Comparison of GRU with **BHP** and the same GRU with the time component removed, retrained on the permuted training set. Statistically significant differences are highlighted and marked with asterisk.

| Dataset | Age | MBD | MIMIC-III | Pendulum | PhysioNet2012 | Retail | Taobao |
| --- | --- | --- | --- | --- | --- | --- | --- |
| Metric | Accuracy | Mean ROC AUC | ROC AUC | $R^2$ | ROC AUC | Accuracy | ROC AUC |
| **Vanilla GRU** | $0.626 \pm 0.004$ | $0.827 \pm 0.001$ | $0.901 \pm 0.002$ | $0.896 \pm 0.010$ | $0.846 \pm 0.004$ | $0.543 \pm 0.002$ | $0.713 \pm 0.004$ |
| **GRU w/o time w/ perm.** | $0.630 \pm 0.004$ | $0.819 \pm 0.001^*$ | $0.890 \pm 0.002^*$ | $0.581 \pm 0.003^*$ | $0.844 \pm 0.005$ | $0.546 \pm 0.003$ | $0.702 \pm 0.006^*$ |

numerical feature before shuffling. For all runs, the last events were kept in their original positions, as some models use the last hidden state in the aggregation step.

Results are presented in Table 3. The Transformer model experienced a significantly small drop due to its attention mechanism. The MLP model did not experience any drop at all because sequence order is inherently not important for aggregation. We observed that while performance dropped for other models, the drop was statistically significant ($p < 0.01$) but less than expected for all real-world datasets. Additionally, the MBD dataset did not experience a significant drop with most methods, suggesting that models do not rely on the order of sequences to make predictions. This indicates that while sequence order is important, it is not as critical for **EvS** assessment of real-world datasets as initially thought. However, we observed that models' performance degraded on the pendulum dataset, indicating that the evaluated models can capture the sequential nature of the data.

**Training on Permuted Sequences**  The second experiment further analyzed datasets to determine if sequential order is important or if sequences can be treated as a **"bag of words."**

We selected the GRU with **BHP** for each dataset, removed the time component from its architecture, and trained it from scratch with both training and test sequences permuted. The results are in Table 4. We observed that for some real-world datasets, the performance drop was not statistically significant. We speculate that such permutation could even serve as a form of data augmentation, since in some cases mean metrics increased with permutation. Notably, after retraining on permuted sequences, we observed a significant drop on the MBD dataset. At first, this seems to contradict the results from the previous section. However, upon considering that the time component was also removed, we conclude that in the MBD dataset, time component is crucial while the order is not.

From both experiments, we conclude that sequence order is important for **EvS** assessment, but it is less critical than expected for real-world datasets and varies from dataset to dataset.

## 4.6 IMPORTANCE OF TIME

Next, we evaluate the role of time in **EvS**. Similarly to the previous section, we perform two experiments: 1) using random time-steps on pre-trained models during testing, and 2) adding or removing time as an extra feature to train the models.

**Incorporation of Event Time Information into Models**  To evaluate the importance of time, we follow a simple procedure. First, we note that time is rescaled during preprocessing. After that, there are three options to incorporate it into the model, all of which are searchable during hyperparameter

Table 5: Including vs. Excluding time as a feature. We take top 3 sets of hyperparameters from **HPO step** for each option and report test metrics. Highlighted bold if adding time significantly improves performance. MLEM not included since it has fixed time process option - copied from best CoLES

| Dataset | Time process | CoLES | GRU | Mamba | MLP | mTAND | PrimeNet | Transformer |
|---|---|---|---|---|---|---|---|---|
| Age | w/o time | $0.632 \pm 0.002$ | $0.622 \pm 0.005$ | $0.612 \pm 0.002$ | $0.587 \pm 0.006$ | $0.583 \pm 0.005$ | $0.582 \pm 0.006$ | $0.609 \pm 0.005$ |
| | with time | $0.633 \pm 0.009$ | $0.629 \pm 0.005$ | $0.616 \pm 0.005$ | $0.588 \pm 0.005$ | $0.588 \pm 0.004$ | $\mathbf{0.594 \pm 0.005}$ | $\mathbf{0.620 \pm 0.005}$ |
| MBD | w/o time | $0.817 \pm 0.002$ | $0.818 \pm 0.001$ | $0.815 \pm 0.001$ | $0.801 \pm 0.002$ | $0.777 \pm 0.013$ | $0.757 \pm 0.011$ | $0.813 \pm 0.001$ |
| | with time | $\mathbf{0.825 \pm 0.000}$ | $\mathbf{0.826 \pm 0.001}$ | $\mathbf{0.822 \pm 0.001}$ | $\mathbf{0.808 \pm 0.000}$ | $\mathbf{0.797 \pm 0.000}$ | $\mathbf{0.781 \pm 0.003}$ | $\mathbf{0.823 \pm 0.000}$ |
| MIMIC-III | w/o time | $0.902 \pm 0.002$ | $0.896 \pm 0.003$ | $0.892 \pm 0.001$ | $0.869 \pm 0.002$ | $0.882 \pm 0.001$ | $0.885 \pm 0.002$ | $0.886 \pm 0.001$ |
| | with time | $0.904 \pm 0.001$ | $0.897 \pm 0.002$ | $\mathbf{0.896 \pm 0.001}$ | $\mathbf{0.879 \pm 0.001}$ | $\mathbf{0.890 \pm 0.005}$ | $0.888 \pm 0.002$ | $\mathbf{0.895 \pm 0.001}$ |
| Pendulum | w/o time | $0.621 \pm 0.003$ | $0.622 \pm 0.007$ | $0.626 \pm 0.004$ | $0.160 \pm 0.000$ | $0.893 \pm 0.019$ | $0.792 \pm 0.010$ | $0.598 \pm 0.003$ |
| | with time | $\mathbf{0.905 \pm 0.002}$ | $\mathbf{0.895 \pm 0.000}$ | $\mathbf{0.908 \pm 0.002}$ | $\mathbf{0.170 \pm 0.000}$ | $\mathbf{0.942 \pm 0.002}$ | $\mathbf{0.852 \pm 0.004}$ | $\mathbf{0.864 \pm 0.003}$ |
| PhysioNet2012 | w/o time | $0.839 \pm 0.002$ | $0.840 \pm 0.003$ | $0.835 \pm 0.001$ | $0.841 \pm 0.002$ | $0.842 \pm 0.002$ | $0.844 \pm 0.001$ | $0.834 \pm 0.004$ |
| | with time | $\mathbf{0.843 \pm 0.002}$ | $0.841 \pm 0.006$ | $\mathbf{0.840 \pm 0.004}$ | $0.837 \pm 0.004$ | $0.845 \pm 0.001$ | $0.842 \pm 0.003$ | $0.838 \pm 0.003$ |
| Taobao | w/o time | $0.705 \pm 0.005$ | $0.685 \pm 0.014$ | $0.693 \pm 0.004$ | $0.637 \pm 0.042$ | $0.664 \pm 0.004$ | $0.653 \pm 0.007$ | $0.702 \pm 0.007$ |
| | with time | $0.712 \pm 0.004$ | $0.705 \pm 0.010$ | $0.666 \pm 0.058$ | $0.666 \pm 0.018$ | $\mathbf{0.679 \pm 0.003}$ | $0.665 \pm 0.040$ | $\mathbf{0.711 \pm 0.002}$ |
| Retail | w/o time | $0.551 \pm 0.001$ | $0.543 \pm 0.001$ | $0.539 \pm 0.001$ | $0.525 \pm 0.000$ | $0.518 \pm 0.000$ | $0.518 \pm 0.001$ | $0.530 \pm 0.002$ |
| | with time | $0.551 \pm 0.001$ | $0.543 \pm 0.001$ | $0.539 \pm 0.001$ | $0.525 \pm 0.002$ | $0.519 \pm 0.001$ | $\mathbf{0.524 \pm 0.004}$ | $\mathbf{0.541 \pm 0.003}$ |

Table 6: Trained models evaluation with random timestamps. Values with statistically significant difference ($p$-value $< 0.01$) in performance are highlighted and marked with asterisk.

| Method | Dataset
Metric
Time | Age
Accuracy | MBD
Mean ROC AUC | MIMIC-III
ROC AUC | Pendulum
$R^2$ | PhysioNet2012
ROC AUC | Retail
Accuracy | Taobao
ROC AUC |
|---|---|---|---|---|---|---|---|---|
| mTAND | Real | $0.582 \pm 0.009$ | $0.798 \pm 0.002$ | $0.888 \pm 0.003$ | $0.941 \pm 0.009$ | $0.841 \pm 0.005$ | $0.519 \pm 0.003$ | $0.672 \pm 0.010$ |
| | Random | $0.581 \pm 0.009$ | $0.795 \pm 0.002^*$ | $0.886 \pm 0.003$ | $0.580 \pm 0.067^*$ | $0.840 \pm 0.005$ | $0.519 \pm 0.004$ | $0.666 \pm 0.010$ |
| PrimeNet | Real | $0.583 \pm 0.011$ | $0.780 \pm 0.006$ | $0.887 \pm 0.004$ | $0.842 \pm 0.017$ | $0.839 \pm 0.004$ | $0.521 \pm 0.003$ | $0.681 \pm 0.010$ |
| | Random | $0.582 \pm 0.010$ | $0.775 \pm 0.006$ | $0.884 \pm 0.004$ | $0.260 \pm 0.108^*$ | $0.840 \pm 0.004$ | $0.521 \pm 0.003$ | $0.680 \pm 0.011$ |

optimization (HPO): **No time** - Do not use time at all; **Time delta** - Compute the time difference from the previous step and concatenate it as a feature; **Absolute time** - Concatenate the rescaled time as a feature.

The results in Table 5 indicate that time significantly improves performance, if added, to three datasets: MBD, MIMIC-III, and Pendulum. Surprisingly, it is important for almost all datasets if we use the Transformer. However, we cannot make the claim for other methods and datasets that the time is not important, as there are various other ways to incorporate it into models that may show statistically significant improvements, but we did not explore them.

**Random Timestamps**  In our work, two methods are specifically designed to model the time component: mTAND (Shukla & Marlin, 2021) and PrimeNet (Chowdhury et al., 2023). We evaluated them on test data with noisy timestamps, where the original timestamps were replaced with random values sorted in ascending order. The results are presented in Table 6. While time is important for these models on the synthetic Pendulum dataset, it did not contribute significantly to the other datasets.

From the observations above, we first see that time is important and contributes to **EvS** assessment. Secondly, we observe that methods specifically designed to work with time do not effectively capture temporal dependencies on real-world datasets. This emphasizes the importance of developing or testing new methods on **EvS** that can model the time component on real-world datasets.

## 5 RELATED WORK

Event Sequences is an important domain that encompasses a variety of tasks. There are several distinct research directions that involve Event Sequences. Predictive Process Monitoring (PPM) is a crucial branch of process mining focused on forecasting the future states of ongoing business processes. It involves analyzing event logs to predict various outcomes, such as process completion time, subsequent events, or final outcomes of process instances Teinemaa et al. (2019); Márquez-Chamorro et al. (2017); Rama-Maneiro et al. (2021); Tax et al. (2020). One related to PPM but distinct problem is Event detection Azib et al. (2023), accurately identifying specific events is vital for making informed decisions. **Deep Learning-based Temporal Point Processes**, which primarily aims at

predicting the type and timing of the next event Xue et al. (2024). **Event Sequence Analysis** employs visual methods for data analysis Zinat et al. (2024). This approach focuses on building libraries. In our work, we focus on another important yet understudied task: Event Sequence assessment, which primarily includes the classification or regression of entire sequences. Therefore, our research is distinct from previously studied benchmarks on time series classification and other studies that also address the domain of **EvS**.

The UCR Time Series Archive, widely used for time-series classification, is limited to univariate time series, offering 128 datasets for algorithm evaluation (Dau et al., 2019). Despite its extensive use, this benchmark does not address the complexity of event sequence data, crucial for many real-world applications. The `torchtime` package (Darke et al., 2022) extends the utility of UEA & UCR datasets by providing reproducible implementations for PyTorch, simplifying data access and ensuring fair model comparisons, it is still primarily focuses on time series classification. EasyTPP (Xue et al., 2024) is a new benchmark targeting streams of discrete events, offering a centralized repository for evaluating TPP models. It emphasizes reproducible research through a standardized benchmarking framework and provides various research assets. However, EasyTPP cannot be extended to handle general EvS, as event sequences generally cannot be modeled using TPP. The sequence of card transactions made by a client is a good example of **EvS**. Each transaction is characterized by attributes such as transaction amount and merchant category code, making them unfit for time series or discrete event streams categories. Authors in Bazarova et al. (2024); Yugay & Zaytsev (2024); Babaev et al. (2022) evaluate several representation learning approaches on event sequences. In Bazarova et al. (2024), the authors propose a protocol for evaluating obtained representations on a set of downstream tasks.

## 6 CONCLUSION

In this work, we presented EBES, an open and comprehensive benchmark for the standardized and transparent comparison of event sequence models. The benchmark includes a diverse range of datasets and models. Additionally, it provides a user-friendly interface and a rich library, allowing for the easy integration of new datasets and the implementation of new models. With these features, EBES has the potential to facilitate future research in event sequence modeling significantly.

We emphasize the importance of **HPO** and cross-validation for fair model evaluation. Moreover, we recommend performing several runs to validate if the model performance is statistically significant, especially on small datasets. This is also supported by scaling experiments, where model rankings tend to change significantly on smaller data sizes and slowly converge to the same point as the data size grows while the standard deviation decreases.

Our analysis of datasets highlights two crucial points. We found that results on the PhysioNet2012 dataset are not statistically distinguishable. Therefore, future researchers should be cautious when deriving conclusions for **EvS** assessment based on results obtained with this dataset. Another observation is that out-of-time data splits naturally tend to have a distribution shift, and one should account for it during model validation and HPO. For example, this appears in the low correlation between the Taobao dataset's validation and test metric values.

We demonstrate that the importance of time and the sequential nature of the data varies for real-world datasets concerning **EvS** assessment. Similarly, different models capture these properties differently. Developing or testing models that inherently account for the time component on real-world data could be a promising direction for future research.

## 7 LIMITATIONS

We acknowledge that conducting a full HPO (Hyperparameter Optimization) process requires substantial computational resources, which may not be available to all users. The development of more efficient strategies for proper model evaluation could be a promising direction for future research.

Our work focuses solely on one task—**EvS** assessment while there are various tasks applied to **EvS**. We leave this for future work.

## 8 REPRODUCIBILITY STATEMENT

We made available all the code necessary to run our experiments and generate the corresponding figures. Additionally, we include raw logs from all experiments, including valuable data obtained

during the HPO process. Each experiment was conducted with fixed random seeds, ensuring that model training yields consistent results when the same seeds are used.

Our code repository includes:

- Configuration files with specifications for the HPO process.
- Implementations of all the methods mentioned in this paper, along with their best hyperparameters.
- A complete data preprocessing pipeline for each dataset used in our study.

By following the instructions provided in our repository, you should be able to reproduce our results accurately.

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

## A  APPENDIX

## B  MODELS DESCRIPTION

**GRU**   We have chosen to use the GRU as one of our base models due to its proven effectiveness in encoding time-ordered sequences Babaev et al. (2022); Rubanova et al. (2019); Tonekaboni et al. (2021); Yoon et al. (2019); Udovichenko et al. (2024). In recent study on neural architecture search Udovichenko et al. (2024)), authors demonstrated that architectures with RNN blocks tend to exhibit higher performance on average on EvS assesment task.

**MLP**    The models applied 3 linear layers with the ReLU nonlinearity and dropouts in between to the aggregated embeddings obtained right after the preprocessing block. So effectively the model is just a basic MLP applied to aggregations. Models for **EvS** handles the sequential nature of data in a special way, ofthen considering the exact time intervals between the events, so we were interested in the performance of the model, that consciously discards the sequential nature of data.

**Mamba**    Mamba Gu & Dao (2023) is a recent state-space model (SSM) that has been designed for efficient handling of complex, long sequences. It incorporates selective state spaces to deliver top-notch performance across different modalities, including language, audio, and genomics, outperforming Transformers in some scenarios. For the best of our knowledge Mamba has not been applied to EvS assessment previously, however, we believe that type of models worth of investigating.

**mTAND**    Authors in Shukla & Marlin (2021) proposed an architecture which learns an embedding of continuous-time values and utilizes an attention mechanism to produce a fixed-length representation of a time series. This procedure is specifically designed to deal with ISTS and has been shown to outperform numerous ordinary differential equations-based models such as Latent ODE and ODE-RNN Rubanova et al. (2019).

**CoLES**    The contrastive pretraining method for sequential data was proposed by Babaev et al. (2022). We specifically focus on this method due to its superior performance compared to other contrastive approaches demonstrated in the work. CoLES learns to encode a sequence into a latent vector by bringing sub-sequences of the same sequence closer in the embedding space while pushing sub-sequences from different sequences further apart.

**PrimeNet**    The method proposed in Chowdhury et al. (2023) also, falls under the category of self-supervised. It utilizes time-sensitive contrastive pretraining and enhances pretraining procedure with data reconstruction tasks to facilitate the usage of unlabeled data. Authors modify mTAN architecture by replacing an RNN block with Feature-Feature Attention.

**MLEM**    The Multimodal Learning Event Model Moskvoretskii et al. (2024) is a recently proposed method for Event Sequences that unifies contrastive learning with generative modeling. It treats generative pre-training and contrastive learning as distinct modalities. First, a contrastive encoder is trained, followed by an encoder-decoder that learns latent states using reconstruction loss while aligning with contrastive embeddings to enhance the embedding information.

## C    DATASETS DESCRIPTION

**PhysioNet2012**    dataset[2] was first intruduced in Goldberger et al. (2000). It includes multivariate time series data with 37 variables gathered from intensive-care unit (ICU) records. Each record contains measurements taken at irregular intervals during the first 48 hours of ICU admission. We used `set-a` as a train set and `set-b` as a test set. Both sets contain 4000 labeled sequences.

**MIMIC-III**    dataset[3] Johnson et al. (2016) consists of multivariate time series data featuring sparse and irregularly sampled physiological signals, collected at Beth Israel Deaconess Medical Center from 2001 to 2012. While we aimed to follow the general pipeline outlined in Shukla & Marlin (2018), we made several modifications to enhance the accuracy and reproducibility of our approach. Importantly, we did not alter the original problem statement: we excluded series that last less than 48 hours and used the first 48 hours of observations from the remaining series to predict in-hospital mortality. These adjustments were necessary to address certain issues and improve the overall robustness of our analysis.

**Age**    dataset[4] consists of 44M anonymized credit card transactions representing 50K individuals. The target is to predict the age group of a cardholder that made the transactions. The multiclass target

---

[2]https://physionet.org/content/challenge-2012/1.0.0/

[3]https://physionet.org/content/mimiciii/1.4/

[4]https://ods.ai/competitions/sberbank-sirius-lesson

label is known only for 30K records, and within this subset the labels are balanced. Each transaction includes the date, type, and amount being charged. The dataset was first introduced in scientific literature in work Babaev et al. (2022).

**Retail**   dataset[5] comprises 45.8M retail purchases from 400K clients, with the aim of predicting a client's age group based on their purchase history. Each purchase record includes details such as time, item category, the cose, and loyalty program points received. The age group information is available for all clients, and the distribution of these groups is balanced across the dataset. The dataset was first introduced in scientific literature in work Babaev et al. (2022).

**MBD**   is a multimodal banking dataset introduced in Dzhambulat et al. (2024). The dataset contains an industrial-scale number of sequences, with data from more than 1.5 million clients. Each client corresponds to a sequence of events. This multi-modal dataset includes card transactions, geo-position events, and embeddings of dialogs with technical support. The goal is to predict the purchases of four banking products in each month, given the historical data from the previous month. For our analysis, we use only card transactions.

Since we focused on the event sequence assessment task, we restricted our setup as follows. To predict the purchases, we use transactions from the preceding month. For example, we use a sequence from June to predict a label by the last day of July. We did not use out-of-time validation, as the labeled time span of the data is less than a year. The authors of the dataset split the data into 5 folds (0–4), we use fold 4 as the test fold.

**Taobao**   dataset comprises user behaviors from Taobao, including clicks, purchases, adding items to the shopping cart, and favoriting items. These events were collected between November 18 and December 15. For our analysis, we treat each week of clicks as a sequence and aim to predict payments for the subsequent 7 days following the selected week. The training set encompasses data from November 18 to December 1, while the test set includes clicks from December 2 to December 15.

**Pendulum**   Inspired by Moskvoretskii et al. (2024) we created a pendulum dataset to evaluate time-dependent models. The Pendulum dataset is specifically designed for event sequence assessment tasks, featuring irregular timestamps and missing values. Its task requires models to consider multiple events for predictions, making it effective in evaluating temporal modelling capabilities.

The dataset simulates damped pendulum motion with varying lengths. Observation times are sampled irregularly using a Hawkes process, emphasizing the importance of accurate event timing for real-world applications. Each sequence in the dataset consists of events represented by time and two normalized coordinates (x, y), with some values randomly dropped. The goal is to predict the damping factor. We publish the reproducible code to generate the dataset.

To model the Hawkes process, we consider the following intensity function $\lambda(t)$ that is given by (1).

$$\lambda(t) = \mu + \sum_{t_i < t} \alpha e^{-\beta(t-t_i)} \tag{1}$$

We used following parameters for the Hawkes process:

- $\mu$ is the base intensity;
- $\alpha$ is the excitation factor, was chosen to be 0.5;
- $\beta$ is the decay factor, was set to 1.
- $t_i$ are the times of previous events before time $t$.

The time points are sampled within the interval $[0, \text{end time}]$, where the end time is sampled from a uniform distribution $U(3, 5)$. To maintain an approximately constant number of points (30) per sequence, we adjust the base intensity $\mu$ as follows:

---

[5]https://ods.ai/competitions/x5-retailhero-uplift-modeling

$$\mu = 30 \times \frac{1 - \alpha}{\text{end time} - 1}$$

This ensures each sequence has a dynamic time interval but approximately the same number of points, preventing the model from learning the timestamp distribution without using timestamp data.

To model the pendulum we consider the second-order differential equation:

$$\theta'' + \left(\frac{b}{m}\right)\theta' + \left(\frac{g}{L}\right)\sin(\theta) = 0 \tag{2}$$

where,

- $\theta''$ is the Angular Acceleration,
- $\theta'$ is the Angular Velocity,
- $\theta$ is the Angular Displacement,
- $b$ is the Damping Factor,
- $g = 9.81 \text{ m/s}^2$ is the acceleration due to gravity,
- $L$ is the Length of pendulum,
- $m$ is the Mass of bob in kg.

To convert this second-order differential equation into two first-order differential equations, we let $\theta_1 = \theta$ and $\theta_2 = \theta'$, which gives us:

$$\theta_2' = \theta'' = -\left(\frac{b}{m}\right)\theta_2 - \left(\frac{g}{L}\right)\sin(\theta_1) \tag{3}$$

$$\theta_1' = \theta_2 \tag{4}$$

Thus, the first-order differential equations for the pendulum simulation are:

$$\theta_2' = -\left(\frac{b}{m}\right)\theta_2 - \left(\frac{g}{L}\right)\sin(\theta_1) \tag{5}$$

$$\theta_1' = \theta_2 \tag{6}$$

In our simulations, the damping factor $b$ is sampled from a uniform distribution $U(1,3)$, and the mass of the bob $m = 1$. The length $L$ of the pendulum is taken from a uniform distribution $U(0.5, 10)$, representing a range of possible lengths from 0.5 to 10 meters. The initial angular displacement $\theta$ is sampled from a uniform distribution $U(0, 2\pi)$, and the initial angular velocity $\theta'$ is sampled from a uniform distribution $U(-\pi, \pi)$, providing a range of initial conditions in radians and radians per second, respectively.

Our primary objective is to predict the damping factor $b$, using the normalized coordinates $x$ and $y$ on the plane. These coordinates are scaled with respect to the pendulum's length, such that the trajectory of the pendulum is represented in a unitless fashion. This normalization allows us to abstract the pendulum's motion from its actual physical dimensions and instead focus on the pattern of movement. Additionally, we randomly drop 10% of values for both coordinates. An illustrative example of this motion is presented in Figure 5.

## D HPO DETAILS

Hyperparameter Optimization (HPO) is a critical step in the development and evaluation of machine learning models. It involves systematically searching for the optimal set of hyperparameters that maximize model performance. In this section, we outline our main evaluation methodology and HPO process, which is detailed in Algorithm 1.

Our approach includes two main steps: the HPO step and the final evaluation step. In the HPO step, we use the Tree-structured Parzen Estimator (TPE) to efficiently search the hyperparameter space.

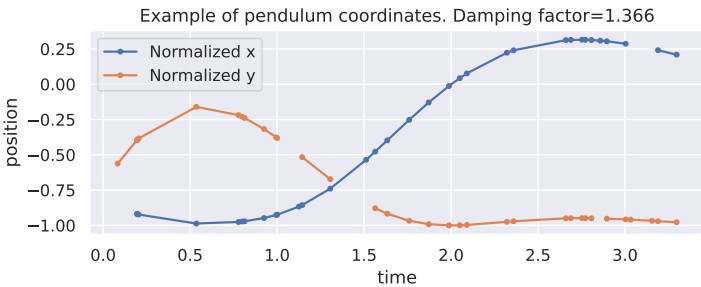

Figure 5: Pendulum motion at various instances, with time steps determined by a Hawkes process.

We split the training dataset into three subsets: `train` (70%), `train-val` (15%), and `hpo-val` (15%). The model is trained on the `train` set, and its performance is evaluated on the `train-val` set to determine when to stop training. The `hpo-val` set is used to update the TPE sampler and guide the selection of hyperparameters.

After the HPO step, we proceed to the final evaluation step. Here, we use the best hyperparameters (BHP) identified in the HPO step to train and evaluate the model multiple times with different random seeds. This ensures that our results are robust and not dependent on a particular random initialization. The training dataset is split into `train` (85%) and `train-val` (15%) sets, and the model is trained until performance on the `train-val` set stops improving or until the training budget is exhausted. Finally, we evaluate the model on the test set and report the mean and standard deviation of the test metrics.

For more details about the HPO process, we refer to our Algorithm 1.

---

**Algorithm 1** Our main evaluation methodolgy and HPO, here $N_{hpo}$ - is HPO budget, $MaxIters$ - training budget, $N_{seeds}$ - a number of iterations for random seed runs.

---
1: $MaxIters = 10^5$
2: $N_{seeds} = 20$
3: start **HPO step**
4: split train dataset randomly into three subsets `train` (70%), `train-val` (15%) and `hpo-val` (15%)
5: initalize TPE
6: **for** $i = 1, 2, \ldots, N_{hpo}$ **do**
7:     set model hyper parameters with TPE
8:     train a model until performance on `train-val` set stops improving or until we run out from the budget $MaxIters$.
9:     update TPE sampler using metrics obtained on `hpo-val`
10: **end for**
11: select best hyper parameters **(BHP)** according to `hpo-val` metrics
12: Start **Final evaluation** step
13: **for** $seed = 1, 2, \ldots, N_{seeds}$ **do**
14:     set a new random $seed$
15:     randomly split train dataset into `train` (85%) and `train-val` (15%) sets
16:     train a model with **BHP** until performance on `train-val` set stops improving or until we run out from the budget $MaxIters$.
17:     evaluate the model on test set
18: **end for**
19: Report $mean$ and $std$ of test metrics from **Final evaluation** step

---

# E SUBSETS METRIC RELATIONSHIPS

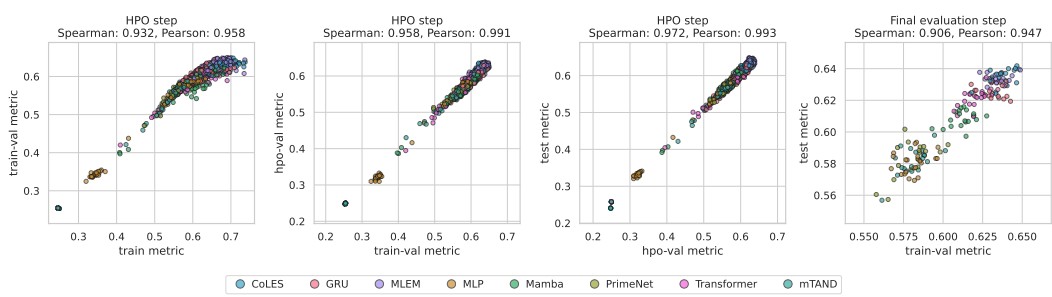

Figure 6: Performance metric relationships and correlations of different subsets among all methods on Age dataset

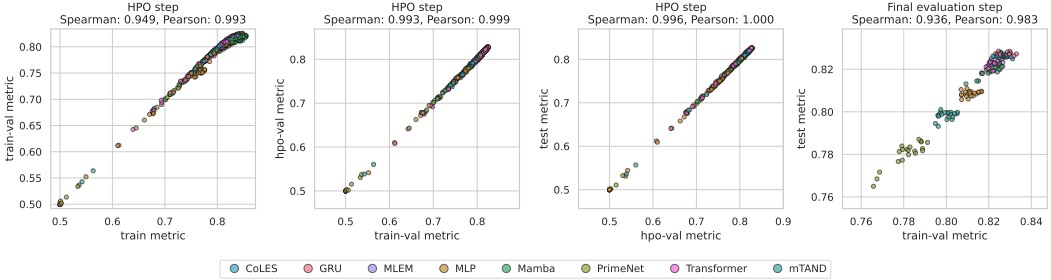

Figure 7: Performance metric relationships and correlations of different subsets among all methods on MBD dataset

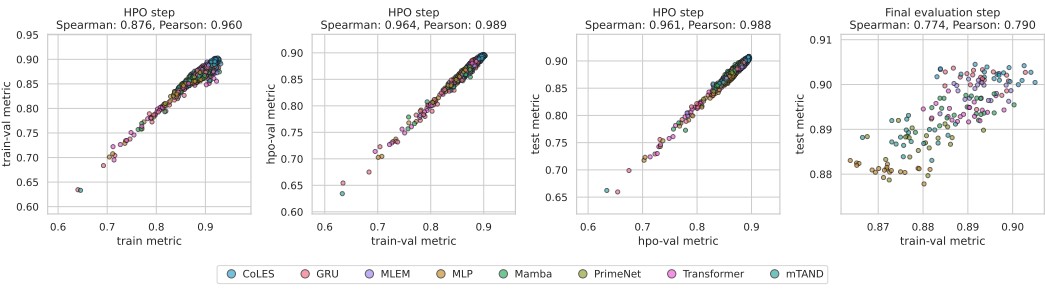

Figure 8: Performance metric relationships and correlations of different subsets among all methods on MIMIC-III dataset

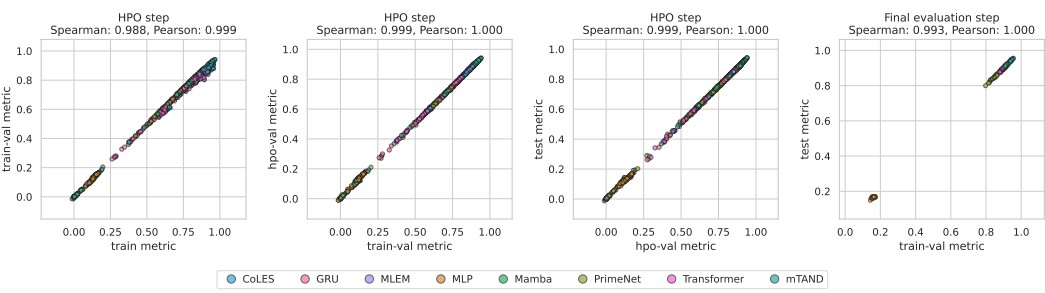

Figure 9: Performance metric relationships and correlations of different subsets among all methods on Pendulum dataset

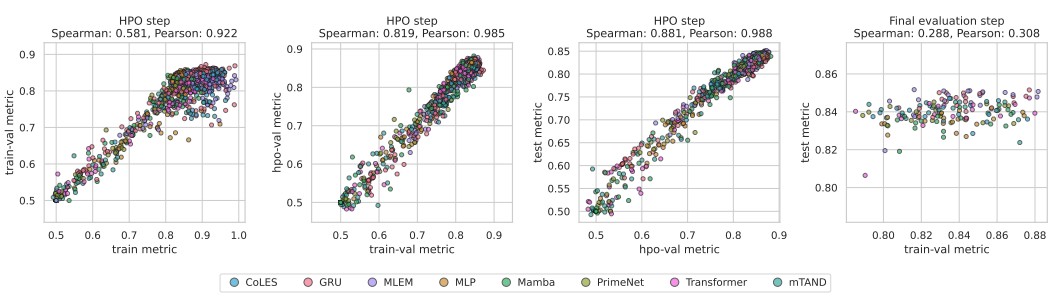

Figure 10: Performance metric relationships and correlations of different subsets among all methods on PhysioNet2012 dataset

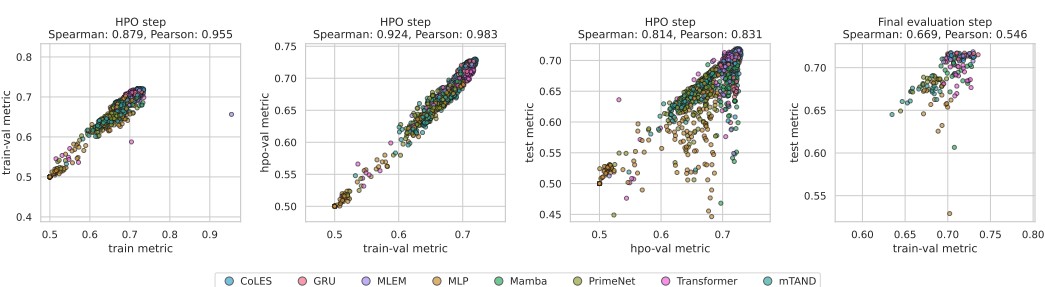

Figure 11: Performance metric relationships and correlations of different subsets among all methods on Taobao dataset

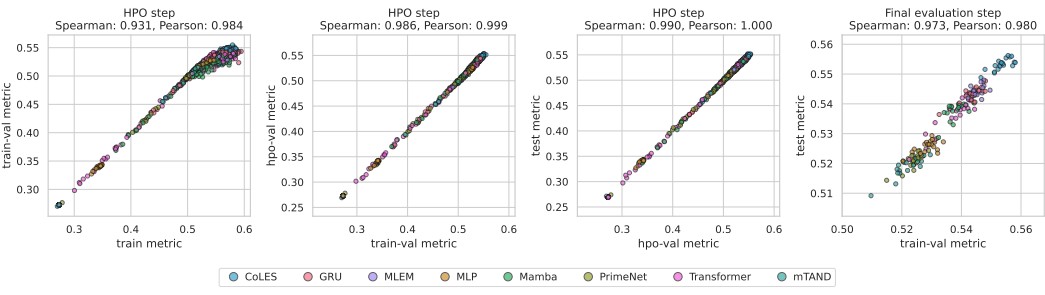

Figure 12: Performance metric relationships and correlations of different subsets among all methods on Retail dataset

## F NUMBER OF LEARNABLE PARAMETERS

We report number of learnable parameters of each model from Table 2 in Table7

| | Age | MBD | MIMIC-III | Pendulum | PhysioNet2012 | Retail | Taobao |
|---|---|---|---|---|---|---|---|
| CoLES | 433,458 | 9,104,269 | 236,481 | 1,065,960 | 401,617 | 2,279,823 | 4,029,658 |
| GRU | 122,326 | 11,224,624 | 1,069,388 | 171,563 | 2,345,326 | 3,810,738 | 103,745 |
| MLEM | 8,969,377 | 14,033,623 | 13,218,268 | 2,601,028 | 1,735,431 | 6,092,930 | 7,412,749 |
| Mamba | 362,712 | 52,396,837 | 798,336 | 2,004,991 | 35,262,580 | 22,422,908 | 1,161,412 |
| Transformer | 1,027,268 | 135,192,356 | 22,169,736 | 3,263,399 | 309,989,960 | 277,133,402 | 1,525,924 |
| mTAND | 27,460 | 5,076,127 | 48,658 | 761,029 | 756,580 | 103,790 | 4,128,148 |
| MLP | 522,440 | 6,425,603 | 1,040,840 | 1,288,793 | 47,242 | 347,269 | 113,566 |
| PrimeNet | 38,840 | 1,661,408 | 1,516,820 | 162,029 | 411,226 | 94,845 | 1,328,836 |

Table 7: Number of learnable parameters in each model from Table 2

## G HPO ANALISYS

This section presents a comprehensive evaluation of different aggregation and normalization approaches, as well as the importance of learning rates, for various models across multiple datasets.

Table 8 compares two aggregation methods: using the last hidden state and the mean of all hidden states. The results indicate that the choice of aggregation method can significantly impact model performance. For instance, in the Age dataset, the mean hidden state approach improves performance for models like GRU and Mamba, while the last hidden state approach is more effective for mTAND. Similarly, Table 9 evaluates the impact of batch normalization on input features. The results show that batch normalization can enhance model performance in many cases.

Additionally, Table 10 ranks the importance of learning rate hyperparameter for different models and datasets using Optuna. The rankings highlight that the learning rate is a critical hyperparameter, with its importance varying across different dataset and model combinations. For example, the learning rate is ranked highest for Mamba across all datasets, indicating its significant impact on model performance. These findings provide valuable insights into the optimal configuration of models for different datasets and can guide future research in hyperparameter optimization.

Table 8: Different aggregation approaches: mean across all hidden states or last hidden state. We take top 3 sets of hyperparameters from **HPO step** for each option and report test metrics. Highlighted bold if adding time significantly improves performance.

| Dataset | Aggregation | CoLES | GRU | Mamba | MLEM | MLP | mTAND | Transformer |
|---|---|---|---|---|---|---|---|---|
| Age | Last hidden | $0.631 \pm 0.001$ | $0.616 \pm 0.004$ | $0.593 \pm 0.003$ | $0.637 \pm 0.004$ | $0.340 \pm 0.002$ | $\mathbf{0.588 \pm 0.004}$ | $0.600 \pm 0.008$ |
| | Mean hidden | $0.630 \pm 0.004$ | $\mathbf{0.629 \pm 0.005}$ | $\mathbf{0.616 \pm 0.005}$ | $0.628 \pm 0.008$ | $\mathbf{0.588 \pm 0.005}$ | $0.579 \pm 0.001$ | $\mathbf{0.617 \pm 0.004}$ |
| MBD | Last hidden | $\mathbf{0.825 \pm 0.000}$ | $\mathbf{0.826 \pm 0.001}$ | $0.822 \pm 0.001$ | $\mathbf{0.823 \pm 0.001}$ | $0.756 \pm 0.000$ | $\mathbf{0.797 \pm 0.000}$ | $0.819 \pm 0.000$ |
| | Mean hidden | $0.821 \pm 0.002$ | $0.822 \pm 0.001$ | $0.822 \pm 0.001$ | $0.820 \pm 0.003$ | $\mathbf{0.808 \pm 0.000}$ | $0.787 \pm 0.001$ | $\mathbf{0.823 \pm 0.000}$ |
| MIMIC-III | Last hidden | $\mathbf{0.904 \pm 0.001}$ | $0.897 \pm 0.002$ | $0.889 \pm 0.001$ | $\mathbf{0.897 \pm 0.001}$ | $0.879 \pm 0.001$ | $0.890 \pm 0.005$ | $\mathbf{0.895 \pm 0.001}$ |
| | Mean hidden | $0.897 \pm 0.002$ | $0.894 \pm 0.002$ | $\mathbf{0.896 \pm 0.001}$ | $0.895 \pm 0.001$ | $0.875 \pm 0.001$ | $0.885 \pm 0.003$ | $0.888 \pm 0.003$ |
| Pendulum | Last hidden | $0.905 \pm 0.002$ | $0.884 \pm 0.003$ | $0.886 \pm 0.007$ | $0.889 \pm 0.001$ | $0.130 \pm 0.001$ | $\mathbf{0.942 \pm 0.002}$ | $0.848 \pm 0.006$ |
| | Mean hidden | $0.903 \pm 0.002$ | $\mathbf{0.895 \pm 0.000}$ | $\mathbf{0.908 \pm 0.002}$ | $0.890 \pm 0.004$ | $\mathbf{0.170 \pm 0.000}$ | $0.899 \pm 0.025$ | $\mathbf{0.864 \pm 0.003}$ |
| PhysioNet2012 | Last hidden | $\mathbf{0.843 \pm 0.002}$ | $0.841 \pm 0.006$ | $\mathbf{0.840 \pm 0.004}$ | $\mathbf{0.844 \pm 0.004}$ | $\mathbf{0.837 \pm 0.004}$ | $0.844 \pm 0.000$ | $0.838 \pm 0.003$ |
| | Mean hidden | $0.827 \pm 0.002$ | $0.803 \pm 0.011$ | $0.806 \pm 0.009$ | $0.826 \pm 0.013$ | $0.808 \pm 0.001$ | $0.844 \pm 0.000$ | $0.831 \pm 0.011$ |
| Taobao | Last hidden | $0.712 \pm 0.004$ | $0.705 \pm 0.010$ | $0.671 \pm 0.038$ | $\mathbf{0.714 \pm 0.002}$ | $0.611 \pm 0.028$ | $0.679 \pm 0.003$ | $0.711 \pm 0.001$ |
| | Mean hidden | $0.698 \pm 0.017$ | $0.696 \pm 0.026$ | $0.666 \pm 0.058$ | $0.707 \pm 0.003$ | $\mathbf{0.666 \pm 0.018}$ | $0.675 \pm 0.002$ | $0.711 \pm 0.002$ |
| Retail | Last hidden | $\mathbf{0.551 \pm 0.001}$ | $\mathbf{0.543 \pm 0.000}$ | $0.528 \pm 0.000$ | $\mathbf{0.545 \pm 0.002}$ | $0.342 \pm 0.001$ | $0.518 \pm 0.001$ | $0.537 \pm 0.001$ |
| | Mean hidden | $0.546 \pm 0.001$ | $0.541 \pm 0.001$ | $\mathbf{0.539 \pm 0.001}$ | $0.540 \pm 0.002$ | $\mathbf{0.525 \pm 0.002}$ | $0.519 \pm 0.001$ | $0.541 \pm 0.003$ |

Table 9: Different normalization approaches: with vs without Batch Normalization for input features. We take top 3 sets of hyperparameters from **HPO step** for each option and report test metrics. Highlighted bold if adding time significantly improves performance.

| Dataset | Normalization | CoLES | GRU | Mamba | MLEM | MLP | mTAND | PrimeNet | Transformer |
|---|---|---|---|---|---|---|---|---|---|
| Age | with norm | 0.629 ± 0.002 | 0.628 ± 0.006 | 0.616 ± 0.005 | 0.636 ± 0.004 | 0.588 ± 0.005 | 0.588 ± 0.004 | 0.584 ± 0.008 | 0.616 ± 0.010 |
| | w/o norm | 0.636 ± 0.005 | 0.624 ± 0.003 | 0.614 ± 0.007 | 0.639 ± 0.006 | 0.582 ± 0.002 | 0.585 ± 0.003 | 0.594 ± 0.005 | 0.617 ± 0.004 |
| MBD | with norm | **0.825 ± 0.000** | **0.826 ± 0.001** | **0.822 ± 0.001** | 0.811 ± 0.010 | **0.808 ± 0.000** | 0.787 ± 0.002 | 0.778 ± 0.003 | 0.822 ± 0.001 |
| | w/o norm | 0.823 ± 0.000 | 0.822 ± 0.001 | 0.819 ± 0.000 | **0.823 ± 0.001** | 0.806 ± 0.001 | **0.797 ± 0.000** | 0.778 ± 0.005 | 0.822 ± 0.001 |
| MIMIC-III | with norm | **0.904 ± 0.001** | **0.897 ± 0.002** | **0.896 ± 0.001** | **0.897 ± 0.001** | **0.879 ± 0.001** | **0.890 ± 0.005** | 0.886 ± 0.000 | **0.895 ± 0.001** |
| | w/o norm | 0.884 ± 0.004 | 0.882 ± 0.006 | 0.880 ± 0.005 | 0.880 ± 0.006 | 0.849 ± 0.008 | 0.877 ± 0.001 | 0.888 ± 0.002 | 0.874 ± 0.001 |
| Pendulum | with norm | 0.872 ± 0.002 | 0.853 ± 0.006 | 0.884 ± 0.003 | 0.844 ± 0.004 | 0.144 ± 0.000 | 0.921 ± 0.001 | 0.829 ± 0.013 | 0.864 ± 0.004 |
| | w/o norm | **0.905 ± 0.002** | **0.895 ± 0.000** | **0.908 ± 0.002** | **0.892 ± 0.002** | **0.170 ± 0.000** | **0.942 ± 0.002** | **0.852 ± 0.004** | 0.859 ± 0.002 |
| PhysioNet2012 | with norm | **0.843 ± 0.002** | **0.841 ± 0.006** | **0.840 ± 0.004** | **0.844 ± 0.004** | **0.837 ± 0.004** | **0.845 ± 0.001** | 0.844 ± 0.001 | **0.838 ± 0.003** |
| | w/o norm | 0.775 ± 0.009 | 0.781 ± 0.012 | 0.832 ± 0.003 | 0.749 ± 0.011 | 0.814 ± 0.009 | 0.808 ± 0.010 | 0.835 ± 0.009 | 0.787 ± 0.006 |
| Taobao | with norm | **0.712 ± 0.004** | 0.705 ± 0.010 | 0.666 ± 0.058 | **0.714 ± 0.002** | 0.666 ± 0.018 | **0.679 ± 0.003** | 0.665 ± 0.040 | **0.711 ± 0.002** |
| | w/o norm | 0.706 ± 0.001 | 0.703 ± 0.006 | 0.685 ± 0.019 | 0.709 ± 0.000 | 0.568 ± 0.068 | 0.654 ± 0.015 | 0.655 ± 0.009 | 0.708 ± 0.001 |
| Retail | with norm | **0.551 ± 0.001** | **0.543 ± 0.000** | **0.539 ± 0.001** | **0.545 ± 0.002** | **0.525 ± 0.002** | **0.519 ± 0.001** | **0.524 ± 0.004** | **0.541 ± 0.003** |
| | w/o norm | 0.539 ± 0.004 | 0.523 ± 0.004 | 0.521 ± 0.003 | 0.530 ± 0.003 | 0.511 ± 0.001 | 0.515 ± 0.003 | 0.518 ± 0.001 | 0.435 ± 0.012 |

Table 10: Learning Rate Importance by Optuna Ranking (Smaller Rank = Higher Importance). There is a unique best Learning Rate for each Dataset/Method combination

| | Age | MBD | MIMIC-III | Pendulum | PhysioNet2012 | Taobao | Retail |
|---|---|---|---|---|---|---|---|
| **CoLES** | 1 | 10 | 2 | 7 | 3 | 1 | 4 |
| **GRU** | 2 | 1 | 2 | 3 | 4 | 1 | 1 |
| **Mamba** | 1 | 1 | 1 | 1 | 1 | 1 | 1 |
| **MLEM** | 2 | 3 | 6 | 11 | 4 | 1 | 1 |
| **MLP** | 2 | 1 | 3 | 5 | 1 | 2 | 1 |
| **mTAND** | 1 | 1 | 1 | 1 | 3 | 1 | 1 |
| **PrimeNet** | 11 | 1 | 1 | 2 | 1 | 1 | 1 |
| **Transformer** | 1 | 4 | 7 | 1 | 10 | 3 | 8 |

