# OpenReview forum: "EBES: Easy Benchmarking for Event Sequences"
_ICLR.cc/2025/Conference — ICLR 2025 Conference Withdrawn Submission_

### Official Review · Reviewer_c1E2 · 2024-11-02

**Soundness:** 2
**Presentation:** 2
**Contribution:** 2
**Rating:** 3
**Confidence:** 4

**Summary:**

This paper presents a new benchmark for classification/regression tasks on event sequences (EBSE). The authors consider different types of event sequences, including numerical and categorical features, as well as varying time granularities. EBSE includes seven datasets, one of which is a synthetic dataset proposed by the authors (Pendulum). After a preprocessing step, various models are tested on these datasets, and the results are presented in the paper. Model performance is evaluated using Monte Carlo cross-validation as proposed by Qing-Song Xu and Yi-Zeng Liang in “Monte Carlo cross validation” (Chemometrics and Intelligent Laboratory Systems, 2001). The corresponding code will be made available by the authors after the paper's acceptance.

**Strengths:**

Benchmarking tools for algorithms used in time series or event sequences are valuable to the scientific community. The authors propose an evaluation protocol that could be of interest. The statistical analysis of the results is also noteworthy.

**Weaknesses:**

In my opinion, this work is at an early stage of development. A significant aspect of this work is not correctly implemented: the evaluation of a model built on event sequences. The splitting of a dataset into training/validation/test sets should be done based on the order (or timestamps, if working with temporal series) of events. If the dataset is split using a random selection of events, the evaluation results may not reflect real performance. The training set must consist of the oldest data, while the test set should contain the most recent data; otherwise, there is a significant risk of overfitting. The authors use a Monte Carlo cross-validation method originally proposed in a chemistry journal (https://libpls.net/publication/MCCV_2001.pdf). This method is not suitable for event sequences or time series, and the original paper does not address this type of data. For a benchmark to be credible, the performance evaluation process of a model must adhere to scientific rigor, and this part needs more detail from the authors. The code is not yet available, and the paper does not explain how the evaluation is performed on event sequences.

The goal of the paper is to present a new benchmark, which should be characterized by at least several aspects: the number of datasets (seven are proposed here), a set of algorithms, the ability to integrate new algorithms easily, a method to build models using these algorithms, and an evaluation procedure and metrics dedicated to event sequence data.

**Questions:**

- Q1. Regarding the weakness mentioned, what is the detailed evaluation process for the performance of a model built on an event sequence?
- Q2. On page 3, the authors state they are focused on “ensuring that the datasets used are high quality and accurately represent the problem domain.” How do you verify that a dataset “accurately represents the problem domain” without involving a domain expert? Additionally, what qualifies as a “high-quality” dataset? The preprocessing step proposed in the benchmark, including the handling of missing values, is also domain-dependent in my opinion. Integrating it into a benchmark can be challenging, although I understand the need to compare different models on the same dataset without missing values.
- Q3. What is an MLP? While the reader may infer it to be a Multilayer Perceptron, the term is not explained, and no references are provided.

---

> ### Author Response · Authors · 2024-11-24
> **Rebuttal by Authors**
>
> We thank the Reviewer's comment and will incorporate clarifications and modifications upon acceptance.  Below we structure our reply referring to mentioned weaknesses (W) and questions (Q) and we are happy for the following  discussion.
>
>
> **Foreword: First and foremost, we would like to make some clarifications, to support our answers below.**
>
> 1. While there are numerous other tasks involving event sequences, such as next event and time prediction, interpolation, anomaly detection, or event detection, we focus on assessing a single attribute of an entire sequence. Our tasks include mainly classification, solely due to the relative ease of finding suitable datasets for this task.
>
> 2. Moreover, we should emphasise some criteria for our datasets selection. When selecting real-world datasets, we focus on event sequence data with irregular time samples, preferably containing both categorical and numerical features. The most distinct feature of our chosen datasets is the presence of a single target per sequence. This target usually represents a characteristic of the sequence.
>
> 3. Finally, it is important to note one of our main contributions is a library for model evaluation and a set of benchmarking protocols.  Our code is designed to facilitate the easy addition of new datasets or methods, allowing for their evaluation in a streamlined manner. In our paper we demonstrated how this tool can be used in order to obtain insides about methods and datasets.
>
> **(W1)The splitting of a dataset into training/validation/test sets should be done based on the order (or timestamps, if working with temporal series) of events.**
>
> We appreciate the reviewer's emphasis on the importance of proper dataset splitting. However, there seems to be a misunderstanding regarding our approach. In our method, each sequence in the dataset is treated as an individual object with a single target value. We do not split the dataset by randomly selecting events. Instead, we split the sequences based on their sequence IDs.
> Regarding temporal splitting, we ensure that test sequences are taken from a later time period whenever possible. This is particularly relevant for the Taobao dataset, where absolute time is available, allowing us to implement this temporal split effectively. This approach helps mitigate the risk of overfitting and ensures that our evaluation results accurately reflect real-world performance.
> For some datasets, all sequences are collected over a short period (less than 2 years). Given the significant seasonality in customer behavior throughout the year, splitting sequences based on time would introduce a significant distribution shift. In practice, with large private datasets, this issue is mitigated by using a large time span that includes more than one period of seasonality. However, most public datasets contain less than one such period. For example, the authors of the MBD [1] dataset propose splitting based on sequences rather than time.
>
> **(W2)The authors use a Monte Carlo cross-validation method. This method is not suitable for event sequences or time series**
>
> Again, we consider each sequence as a separate object, which makes Monte Carlo cross-validation suitable. We clarified the description of the model performance evaluation in the subsection 2.2 (``model evaluation’’ paragraph).
>
> **(W3)The code is not yet available, and the paper does not explain how the evaluation is performed on event sequences.**
>
> The code is actually available in the supplementary materials. We will ensure that the paper clearly explains how the evaluation is
> performed on event sequences to avoid any confusion.
>
> **(W4)The goal of the paper is to present a new benchmark, which should be characterized by at least several aspects: the number of datasets (seven are proposed here), a set of algorithms, the ability to integrate new algorithms easily, a method to build models using these algorithms, and an evaluation procedure and metrics dedicated to event sequence data.**
>
> We appreciate the reviewer's feedback. Our paper indeed aims to present a new benchmark for event sequence assessment, and we have addressed all the aspects mentioned.

---

> ### Author Response · Authors · 2024-11-24
> **Continuation of Rebuttal by Authors**
>
> **(Q1)What is the detailed evaluation process for the performance of a model built on an event sequence?**
>
> We clarified the description of the model performance evaluation in the subsection 2.2 (``model evaluation’’ paragraph).
>
>
> **Revised text:**
> ```
> In our procedure, we first conduct an extensive hyperparameter search. Randomness can destabilize models, causing large variances in results across training runs. Ignoring this sensitivity can create a false perception of research progress. Therefore, after determining the optimal hyperparameters, we perform Monte Carlo cross-validation (MCCV) with 20 seeds. At each MCCV step we train the model with the best hyperparameters and pick the checkpoint based on the (randomly sampled) validation set. Finally, the checkpoint is evaluated on the held-out test set. The mean test score across all seeds is reported as the model performance.
> ```
> **(Q2)What qualifies as a “high-quality” dataset?**
>
> Under the term "dataset quality," we are not referring to the accuracy or correctness of the data itself. Instead, we are discussing the quality from a benchmarking perspective. The crucial property for a dataset to be included in a benchmark is its ability to discriminate between models. This means we should be able to determine if one method performs statistically significantly better than another on the dataset's problem. If the performance differences between methods are insignificant on a particular dataset, its inclusion in the benchmark becomes questionable.
>
> **(Q3)What is an MLP?**
>
> We provide description of all the models in the appendix:
> ```
> \paragraph{MLP}
> The models applied 3 linear layers with the ReLU nonlinearity and dropouts in between to the aggregated embeddings obtained right after the preprocessing block.
> So effectively the model is just a basic MLP applied to aggregations.
> Models for \ES handles the sequential nature of data in a special way, ofthen considering the exact time intervals between the events, so we were interested in the performance of the model, that consciously discards the sequential nature of data.
>
> ```
> [1] Dzhambulat, Mollaev, et al. "Multimodal Banking Dataset: Understanding Client Needs through Event Sequences." arXiv preprint arXiv:2409.17587 (2024).
>
> **Overall, we thank the reviewer once again for their valuable feedback and hope that we have successfully addressed all the issues raised. If the reviewer feels that our paper now merits a stronger chance for publication, we would be extremely grateful for a score increase. Your constructive input has been instrumental in improving the quality of our work.**

---

### Official Review · Reviewer_bKZ8 · 2024-11-02

**Soundness:** 3
**Presentation:** 2
**Contribution:** 2
**Rating:** 5
**Confidence:** 4

**Summary:**

The paper introduces EBES, a benchmarking framework designed specifically for evaluating event-sequence (EvS) data, where sequences of events are characterized by irregular time intervals and a mix of categorical and numerical features. The paper proposes a unified framework including datasets, models, evaluation protocols to enable benchmarking on event-sequence tasks.

**Strengths:**

EBES provides a unified framework including both datasets, models, and experimental protocols to facilitate reproducible research and consistent evaluations. EBES includes datasets from different application domains, such as medical records, transaction sequences, and synthetic datasets, enhancing its diversity, applicability and utility. The framework also offers a wide range of comprehensive models, such as GRU, Transformer, or specialized ones, mTAND and CoLES. This ensures a thorough evaluation about model performance on event sequences related tasks. The framework also analyzes the importance of modeling time and sequence order in event-sequence data in prediction accuracy.

**Weaknesses:**

The proposed benchmark framework lacks a comparison with existing tools such as the multi-level task framework for event sequences [1] or [2], hence causes difficulties to assert its advantages over existing frameworks and tools. The dataset diversity is also limited. It is suggested that authors should extend their framework to other application domains, such as the smart city, weather and renewable energy, traffic, social network to further improve the applicability of their framework. Further, evaluation on large scale datasets or real-time event datasets should be conducted to assert the performance and scalability of the benchmark.

[1] https://arxiv.org/abs/2408.04752
[2] https://arxiv.org/pdf/2310.16485

**Questions:**

- How does EBES compare to other event-sequence benchmarks and frameworks, such as the multi-level task framework [1] or other existing benchmarks in event-sequence modeling and temporal point processes?
- Does EBES have plans to incorporate additional datasets from other domains like smart cities, weather forecasting, renewable energy, traffic data, or social networks? Are there plans for incorporating such datasets?
- Has EBES been tested with large-scale or real-time datasets to evaluate its scalability and performance under real world conditions?

---

> ### Author Response · Authors · 2024-11-24
> **Rebuttal by Authors**
>
> We thank the Reviewer's comment and will incorporate clarifications and modifications upon acceptance.  Below we structure our reply referring to mentioned weaknesses (W) and questions (Q) and we are happy for the following  discussion.
>
>
> **Foreword: First and foremost, we would like to make some clarifications, to support our answers below.**
>
> 1. While there are numerous other tasks involving event sequences, such as next event and time prediction, interpolation, anomaly detection, or event detection, we focus on assessing a single attribute of an entire sequence. Our tasks include mainly classification, solely due to the relative ease of finding suitable datasets for this task.
>
> 2. Moreover, we should emphasise some criteria for our datasets selection. When selecting real-world datasets, we focus on event sequence data with irregular time samples, preferably containing both categorical and numerical features. The most distinct feature of our chosen datasets is the presence of a single target per sequence. This target usually represents a characteristic of the sequence.
>
> 3. Finally, it is important to note one of our main contributions is a library for model evaluation and a set of benchmarking protocols.  Our code is designed to facilitate the easy addition of new datasets or methods, allowing for their evaluation in a streamlined manner. In our paper we demonstrated how this tool can be used in order to obtain insides about methods and datasets.
>
>
>
> **(W1) The proposed benchmark framework lacks a comparison with existing tools such as the multi-level task framework for event sequences [1] or [2], hence causes difficulties to assert its advantages over existing frameworks and tools.**
>
> [1] https://arxiv.org/abs/2408.04752
> [2] https://arxiv.org/pdf/2310.16485
>
>
> We carefully evaluated both papers. The first paper proposed a framework for event sequence analytics. The second paper focuses on event detection and uses a Credit Fraud dataset from 2014. While this approach can be used for event sequence classification, the primary goal of this work, as cited, is "not to attain state-of-the-art metrics for each use case presented here but to demonstrate our method’s versatility and the easy use of the package in various domains." Since we evaluate models and its different components, a straightforward comparison is beyond the scope of our work. However, both papers highlight the importance of event sequences for various applications, and we have included both in our literature review.
>
>
> Revised part of the text, which includes references:
> ```
> Event Sequences is an important domain that encompasses a variety of tasks. There are several distinct research directions that involve Event Sequences. Predictive Process Monitoring is a crucial branch of process mining focused on forecasting the future states of ongoing business processes. It involves analyzing event logs to predict various outcomes, such as process completion time, subsequent events, or final outcomes of process instances~\cite{teinemaa2019outcome, marquez2017predictive, rama2021deep, tax2020interdisciplinary}.
> \textbf{Deep Learning-based Temporal Point Processes}, which primarily aims at predicting the type and timing of the next event~\cite{xue2024easytpp}.
> \textbf{Event Sequence Analysis} employs visual methods for data analysis~\cite{zinat2024multi}. This approach focuses on building libraries. In our work, we focus on another important yet understudied task: Event Sequence assessment, which primarily includes the classification or regression of entire sequences. Therefore, our research is distinct from previously studied benchmarks on time series classification and other studies that also address the domain of \ES.
> ```
> **(Q1) How does EBES compare to other event-sequence benchmarks and frameworks, such as the multi-level task framework [1] or other existing benchmarks in event-sequence modelling and temporal point processes?**
>
> Our work and [1] study the same domain of Event Sequences but we focus on different problems as mentioned in Foreword.

---

> > ### Comment · Reviewer_bKZ8 · 2024-11-26
> >
> > While I appreciate the author's responses and clarifications, I still have reservations about the paper's contributions. As the authors noted, EBES is specifically designed for classification tasks, and the datasets included remain limited to a few application domains. Therefore, I believe the framework requires further development to reach a more mature state suitable for publication. I look forward to seeing the progress of this work in future submissions.

---

> > > ### Author Response · Authors · 2024-12-02
> > > **Additional clarification**
> > >
> > > We disagree with the notion that the merit of our work should be evaluated solely based on the number of datasets used.
> > >
> > > Our contribution is that it is the first attempt in the literature to propose a benchmark for this specific domain and task. It is crucial to differentiate between the evaluation framework (including code and evaluation protocols), which is introduced for the first time, and the list of datasets used.
> > >
> > > Furthermore, it is inaccurate to assert that our framework is developed solely for classification tasks. While it is true that our dataset selection leans more towards classification tasks, this bias is reasonable given the greater availability of open classification datasets. Therefore, it is important to separate the framework from the datasets used. Our framework is designed to accommodate the addition of new datasets and supports both regression and classification tasks.
> > >
> > > Moreover, we invite the reviewer, to propose new open datasets to include and we gladly will look into them if they fit our problem and the task.

---

> ### Author Response · Authors · 2024-11-24
> **Continuation of Rebuttal by Authors**
>
> **(Q2) Does EBES have plans to incorporate additional datasets from other domains like smart cities, weather forecasting, renewable energy, traffic data, or social networks? Are there plans for incorporating such datasets?**
>
> Yes, we want to and we plan to extend our benchmark with new datasets from mentioned areas and possibly more. However, it is not that easy to find open datasets which meet our criteria mentioned in the foreword. While many papers on temporal point processes also study event sequences, many of these datasets do not have the target associated with the entire sequence.
>
> Currently, we are already adding the MIMIC-III dataset with a regression target and new dataset from [1] . We will update reviews as soon as possible to get the first results. The final results for all the models will be ready for the camera ready version.
> To further highlight challenges in dataset curation. During the rebuttal period we already evaluated the dataset from [2]. While it fits our domain the dataset is “relatively simple” and we get 0.97 ROC AUC with RNN model. So, we are not going to include it since it does not help to differentiate models.
>
> [1] https://arxiv.org/pdf/2401.03246
>
> [2] https://decan.lexpage.net/files/MSR-2023.pdf
>
>
> **(Q3) Has EBES been tested with large-scale or real-time datasets to evaluate its scalability and performance under real world conditions?**
>
> Yes, we evaluated EBES on a real world dataset, for example MDB [4] is currently one of the largest event sequence datasets with  1.5 million users. This dataset is an anonymized version of real banking translational data. According to  real-time datasets, there is maybe some confusion from our side, as we understand it is about real-time data obtained via API, please correct us if we are wrong. If it is the case, then  this data type is suitable indeed and it is possible to collect such datasets. However, we prioritise datasets at least briefly mentioned in the literature because it is more transparent.
>
>
> **Overall, we thank the reviewer once again for their valuable feedback and hope that we have successfully addressed all the issues raised. If the reviewer feels that our paper now merits a stronger chance for publication, we would be extremely grateful for a score increase. Your constructive input has been instrumental in improving the quality of our work.**

---

### Official Review · Reviewer_pdqo · 2024-11-03

**Soundness:** 3
**Presentation:** 2
**Contribution:** 2
**Rating:** 3
**Confidence:** 4

**Summary:**

This paper proposes a benchmark for event sequence classification and regression. The benchmark includes several commonly used datasets and a synthetic dataset. It examines the behaviors of several machine learning models and how they capture time and orders related features of events.

**Strengths:**

This paper provides experimental results of assessing several sequence models using different types of datasets, which could benefit future research in this field.

This paper adopts the hyperparameter optimization and Monte-Carlos cross validation for a fair comparison among models.

**Weaknesses:**

Even though the paper compares time series data and event sequences in Figure 1, it fails to explain the differences in terms of research challenges of the prediction tasks, thus justifying the motivation and novelty of the proposed benchmark. In fact, several models selected in the paper to assess are the ones for time series data, not particularly for event sequences.

The selection of the publicly available datasets does not show much diversity in terms of data scales, formats, difficulty levels (such as the number of classes), and domains. The assessment results also indicate that the models tend to behave similarly in different datasets. Therefore, it is hard to identify the strengths and weaknesses of a model using the benchmark datasets. The paper also generates a synthetic dataset, Pendulum. But it is unclear how the synthetic dataset complements those real-world datasets and what unique perspectives it can bring to the assessment, especially since it is the only dataset of regression tasks and could introduce biases to the assessment. More datasets for regression tasks, especially the real world ones, should be included in the benchmark.

**Questions:**

What are the unique research challenges for event sequence prediction compared to general sequence prediction? The experiment results indicate that general sequence prediction models can also perform well for event prediction. Why is there a need to propose a new benchmark? What are the unique benefits of the new benchmark compared to the existing ones?

What are the principles of selecting the real-world datasets (besides they are publicly available) and designing the synthetic datasets? How can those datasets assess the behavior of a model from different perspectives?

The selected MLP, GRU, and Transforms are general DL models which can be used to build more specific sequence models for a certain application domain. The paper should include more specific models that better represent the SOTA for event prediction.

The benchmark mainly focuses on sequence classification with a small number of classes. In those tasks, time/order could help the performance but not critical as it is not part of the predicted results. The benchmark could include more complex and challenging prediction tasks, such as those mentioned in the first paragraph in Page 2, to have a more thorough study on event sequence models.

The results of Table 2 and Figure 3 need to be explained. For example, why GRU outperforms Transformer even though the latter is more complex? What is the scale of each model in terms of the number of the tunable parameters, what conclusion can be drawn from Table 2, how pretrained model performs differently than others, why mTAND performs better than others with the Pendulum dataset but fails to do so with other datasets. In Figure 3, if no linear trend is observed, what does it suggest?

There is no description of the Taobao dataset used in the assessment.

---

> ### Author Response · Authors · 2024-11-24
> **Rebuttal by Authors**
>
> We thank the Reviewer's comment and will incorporate clarifications and modifications upon acceptance.  Below we structure our reply referring to mentioned weaknesses (W) and questions (Q) and we are happy for the following  discussion.
>
>
> **Foreword: First and foremost, we would like to make some clarifications, to support our answers below.**
>
> 1. While there are numerous other tasks involving event sequences, such as next event and time prediction, interpolation, anomaly detection, or event detection, we focus on assessing a single attribute of an entire sequence.  Our tasks include mainly classification, solely due to the relative ease of finding suitable datasets for this task.
>
> 2. Moreover, we should emphasise some criteria for our datasets selection. When selecting real-world datasets, we focus on event sequence data with irregular time samples, preferably containing both categorical and numerical features. The most distinct feature of our chosen datasets is the presence of a single target per sequence. This target usually represents a characteristic of the sequence.
>
> 3. Finally, it is important to note one of our main contributions is a library for model evaluation and a set of benchmarking protocols.  Our code is designed to facilitate the easy addition of new datasets or methods, allowing for their evaluation in a streamlined manner. In our paper we demonstrated how this tool can be used in order to obtain insides about methods and datasets.
>
> **(W1)  Even though the paper compares time series data and event sequences in Figure 1, it fails to explain the differences in terms of research challenges of the prediction tasks, thus justifying the motivation and novelty of the proposed benchmark. In fact, several models selected in the paper to assess are the ones for time series data, not particularly for event sequences.**
>
>
> We appreciate the reviewer's feedback regarding the need for a clearer explanation of the research challenges specific to event sequences. While some models selected for assessment are traditionally used for time series data, their performance on event sequences highlights the unique challenges and justifies the need for our proposed benchmark. For instance, models like MTAND and Transformer, which perform well in their respective fields, did not rank as highly in our event sequence tasks, and perform worse than GRU. This discrepancy underscores the distinct nature of event sequence assessment and the necessity for specialised approaches, thereby validating the novelty and motivation of our benchmark.
>
>
>
> **(W2) The selection of the publicly available datasets does not show much diversity in terms of data scales, formats, difficulty levels (such as the number of classes), and domains. The assessment results also indicate that the models tend to behave similarly in different datasets. Therefore, it is hard to identify the strengths and weaknesses of a model using the benchmark datasets.**
>
> Our selection of publicly available datasets actually offers a significant degree of diversity. The data scales range widely, from 4k sequences in Physionet2012 to 7.4 million in MBD. The domains covered are equally varied, including E-commerce, Retail, transactions, medical data, and synthetic physical datasets. Additionally, the difficulty levels vary across datasets, as evidenced by the different performance metrics achieved by the models on each dataset. Furthermore, the observation that models tend to behave similarly across different datasets aligns with our goal of establishing a robust method ranking. This consistency might indicate that event sequence prediction is a well-defined field where top-performing methods exhibit similar behaviors.
>
> Moreover, we are adding new dataset from [1], since all the revierews raised the concerns about lack of the datasets. We will update reviews as soon as possible we get the first results. The final results for all the models will be ready for camera ready version.
>
> [1] https://arxiv.org/pdf/2401.03246

---

> ### Author Response · Authors · 2024-11-24
> **Continuation of Rebuttal by Authors**
>
> **(W3) The paper also generates a synthetic dataset, Pendulum. But it is unclear how the synthetic dataset complements those real-world datasets and what unique perspectives it can bring to the assessment, especially since it is the only dataset of regression tasks and could introduce biases to the assessment. More datasets for regression tasks, especially the real world ones, should be included in the benchmark.**
>
> The Pendulum dataset is specifically designed for event sequence assessment tasks, featuring irregular timestamps and missing values that align it with our desired category. Its task requires models to consider multiple events for predictions, making it effective in evaluating temporal modelling capabilities. As seen in Table 2, while MLP performs adequately on most real-world datasets by using aggregated sequences and disregarding temporal structure, it struggles on the Pendulum dataset. This highlights Pendulum's effectiveness in assessing temporal dependencies. For instance, MTAND, with its time-focused attention mechanism, performs best on Pendulum, whereas MLP performs poorly. To further enrich our benchmark, we are already adding the MIMIC-III dataset with a regression target, and we will consider additional real-world regression datasets to complement Pendulum and provide a more comprehensive assessment.
>
> Revised part of the text:
> ```
> \paragraph{Pendulum} \label{sec:pendulum}
> Inspired by ~\cite{moskvoretskii2024selfsupervised} we created a pendulum dataset to evaluate time-dependent models.
> The Pendulum dataset is specifically designed for event sequence assessment tasks, featuring irregular timestamps and missing values. Its task requires models to consider multiple events for predictions, making it effective in evaluating temporal modelling capabilities.
> ```
>
>
> **(Q1) What are the unique research challenges for event sequence prediction compared to general sequence prediction? The experiment results indicate that general sequence prediction models can also perform well for event prediction. Why is there a need to propose a new benchmark? What are the unique benefits of the new benchmark compared to the existing ones?**
>
> In time series (both regular and irregular), events that occur in close temporal proximity often carry related information and are typically interconnected. However, in our context, events can be significantly separated in time and may be independent of each other. On the contrary, classification of regular time series is effectively performed with convolutional layers, which take into account relation to neighboring time stamps [2].  Our benchmark reveals that models which work well with RSTS and ISTS, such as MTAND and Transformer, do not rank as highly on event sequence assessment datasets, with GRU performing better. This challenges the stereotype that Transformer models are universally superior to RNNs, and highlights the need for a separate benchmark tailored to event sequence assessment. A key benefit of our benchmark is its emphasis on rigorous hyperparameter optimization (HPO), which is crucial for fair and thorough model evaluation. We provide a high-quality code base specifically designed for conducting such evaluations. Additionally, our benchmark analyses the importance of modeling time and event order. This addresses significant gaps in existing assessments and underscores the need for our specialized benchmark.
>
> [2] https://link.springer.com/article/10.1007/s10618-020-00727-3

---

> > ### Comment · Reviewer_pdqo · 2024-12-02
> >
> > >In time series (both regular and irregular), events that occur in close temporal proximity often carry related information and are typically interconnected. However, in our context, events can be significantly separated in time and may be independent of each other. On the contrary, classification of regular time series is effectively performed with convolutional layers, which take into account relation to neighboring time stamps [2].
> >
> > If the main difference lies in the regularity of time series, is there a proof that the selected datasets are significantly more irregular than the ones in current benchmarks? And if so, the irregularity can also be used as one of the characteristics of a dataset and it would be interesting to see its impact on model performance.

---

> ### Author Response · Authors · 2024-11-24
> **Continuation of Rebuttal by Authors**
>
> **(Q2) What are the principles of selecting the real-world datasets (besides they are publicly available) and designing the synthetic datasets? How can those datasets assess the behavior of a model from different perspectives?**
>
> When selecting real-world datasets, we focused on event sequence data with irregular time samples, preferably containing both categorical and numerical features. The most distinct feature of our chosen datasets is the presence of a single target per sequence. This target usually represents a characteristic of the sequence. The motivation behind the Pendulum dataset is described in answer W3. For the real-world datasets, our goal was to cover as many different fields as possible to assess model behaviour from various perspectives, including e-commerce, retail, transactions, medical data, and synthetic physical datasets. This diversity ensures that our benchmark evaluates models across a range of scenarios and challenges, providing a comprehensive assessment of their capabilities.
> It is challenging to pinpoint the exact components and strengths evaluated by each dataset, but we can make informed speculations. The Age and Retail datasets likely emphasize the ability to make predictions based on a set of events, with less focus on temporal modeling. In contrast, the Pendulum dataset requires careful temporal modeling and, as an ISTS dataset, models designed for this type of data should perform better on it. The MBD dataset, being the largest in this field, allows us to observe the scaling properties of models. The Physionet2012 dataset may not be ideal for assessing performance in event sequence prediction. The MIMIC-III dataset evaluates a model's ability to handle complex medical data, while the Taobao dataset assesses performance on high-frequency user-behavior data. By including these diverse datasets, our benchmark ensures a comprehensive evaluation of models from various perspectives.
>
> **(Q3) The selected MLP, GRU, and Transforms are general DL models which can be used to build more specific sequence models for a certain application domain. The paper should include more specific models that better represent the SOTA for event prediction.**
>
>
> Indeed, the paper could benefit from including more specific models to better define the state-of-the-art (SOTA) for event sequence assessment. We included as many available SOTA models as we could reproduce. One of the motivations for creating this benchmark is the lack of research in this field, which necessitated adapting models from other domains. MTAND and PrimeNet are SOTA for classification in irregularly sampled time series, while CoLES is a SOTA for event sequence assessment. GRU and Transformer serve as foundational models for these methods, and we evaluated them without modifications to understand their baseline performance. MLP was included as a baseline that does not model sequential structure at all. Our benchmark aims to fill the gap in research and provide a comprehensive evaluation of both foundational and specialized models for event sequence assessment.
>
>
> **(Q4) The benchmark mainly focuses on sequence classification with a small number of classes. In those tasks, time/order could help the performance but not critical as it is not part of the predicted results. The benchmark could include more complex and challenging prediction tasks, such as those mentioned in the first paragraph in Page 2, to have a more thorough study on event sequence models.**
>
>
> Many of the tasks described on Page 2 have already been thoroughly studied. For instance, next event prediction tasks are typically covered by Temporal Point Process (TPP) models, and EasyTPP offers a centralized repository for evaluating TPP models. Tasks like interpolation, extrapolation, and forecasting are usually relevant to Irregularly Sampled Time Series (ISTS) or Regularly Sampled Time Series (RSTS). ContiFormer and other works cover many ISTS methods and compare them on these tasks. Our benchmark specifically focuses on event sequence assessment, where time and order can still significantly impact performance. While these tasks may not be as complex as those mentioned, they provide a foundational evaluation for event sequence models. We believe that adding more tasks to this benchmark would not be beneficial, as the setting of event sequence assessment is under-researched. Mixing it with other well-researched topics could dilute the focus and hinder the development of specialized methods for this specific area.

---

> > ### Comment · Reviewer_pdqo · 2024-12-02
> >
> > > While these tasks may not be as complex as those mentioned, they provide a foundational evaluation for event sequence models.
> >
> > I still think the technical contribution of the paper is limited since the selected tasks (classification and regression) and models do not go significantly beyond the traditional time series data analysis.

---

> ### Author Response · Authors · 2024-11-24
> **Continuation of Rebuttal by Authors**
>
> **(Q5) The results of Table 2 and Figure 3 need to be explained. For example, why GRU outperforms Transformer even though the latter is more complex? What is the scale of each model in terms of the number of the tunable parameters, what conclusion can be drawn from Table 2.**
>
> Observation that GRU outperforms Transformer aligns with other works, such as CoLES (Section 4.2.1, Table 3), which also shows that Transformer underperforms as an encoder for event sequence assessment compared to GRU. This suggests that the added complexity of Transformer does not necessarily translate to better performance in this specific context.
>
> To provide further insight, we present the number of learnable parameters for each model in the table below:
>
>
>
> | Model       | Age      | MBD        | MIMIC-III  | Pendulum   | PhysioNet2012 | Retail    | Taobao    |
> |-------------|----------|------------|------------|------------|---------------|-----------|-----------|
> | **CoLES**   | 433,458  | 9,104,269  | 236,481    | 1,065,960  | 401,617      | 2,279,823 | 4,029,658 |
> | **GRU**     | 122,326  | 11,224,624 | 1,069,388  | 171,563    | 2,345,326    | 3,810,738 | 103,745   |
> | **MLEM**    | 8,969,377| 14,033,623 | 13,218,268 | 2,601,028  | 1,735,431    | 6,092,930 | 7,412,749 |
> | **Mamba**   | 362,712  | 52,396,837 | 798,336    | 2,004,991  | 35,262,580   | 22,422,908| 1,161,412 |
> | **Transformer** | 1,027,268 | 135,192,356| 22,169,736 | 3,263,399  | 309,989,960 | 277,133,402| 1,525,924 |
> | **mTAND**   | 27,460   | 5,076,127  | 48,658     | 761,029    | 756,580      | 103,790   | 4,128,148 |
> | **MLP**     | 522,440  | 6,425,603  | 1,040,840  | 1,288,793  | 47,242       | 347,269   | 113,566   |
> | **PrimeNet**| 38,840   | 1,661,408  | 1,516,820  | 162,029    | 411,226      | 94,845    | 1,328,836 |
>
> **Table:** Number of learnable parameters in each model from Table 2
>
>
>
> The table illustrates that the Transformer model has a significantly higher number of parameters yet is outperformed by GRU. We have included this table in the Appendix.
>
> **(Q6) how pretrained model performs differently than others**
> The performance of pretrained models, such as CoLES, varies depending on the nature of the task. CoLES pretraining suggests that it should improve GRU metrics on tasks where the target is a characteristic of an observed sequence, such as Age, Pendulum, and Retail. However, on datasets where the target is somehow connected to future events, such as Taobao, MIMIC-III, MBD, and PhysioNet, the pretraining does not provide a significant boost. Additionally, MLEM performs similarly to CoLES, likely due to its usage of pretrained CoLES components
>
> **(Q7) why mTAND performs better than others with the Pendulum dataset but fails to do so with other datasets**
>
> MTAND was specifically designed for modeling Irregularly Sampled Time Series (ISTS), and its internal structure assumes such relationships in the data. The Pendulum dataset, with its numerical features sampled irregularly, happens to be a perfect example of such a dataset. This alignment explains why MTAND performs better on the Pendulum dataset compared to others. This also suggests that MTAND is a strong option for tasks requiring careful temporal modeling. However, its performance does not translate as well to other datasets, indicating that its strengths are particularly suited to scenarios with irregular sampling
>
> **Revised part of the text containing conclusions:**
> ```
> In this section, we address the main question of the benchmark: \textbf{Which model performs the best?}
> The results are presented in Table~\ref{tab:main}, where methods are sorted from top to least performing.
> Along with the mean performance we report method's rank as a superscript.
> We performed pairwise Mann--Whitney $U$ test~\citep{mann1947test} with Benjamini--Hochberg correction~\citep{benjamini1995controlling},  methods with no significant performance difference ($p > 0.01$) share the same superscript.
> All top three performing methods are based on GRU with different pre-training strategies. CoLES improves metrics on tasks where the target is a characteristic of an observed sequence, such as Age, Pendulum, and Retail. However, on datasets where the target is somehow connected to future events, such as Taobao, MIMIC-III, MBD, and PhysioNet, the pretraining does not provide a significant boost. MLEM performs similarly to CoLES, likely due to its usage of pretrained CoLES components.
>
> Transformer and Mamba comes next in rating, suggesting that this architectures are less suitable for \ES assessment. mTAND~\citep{shukla2021multitime} excelled on the Pendulum dataset due to its architecture tailored for modeling the time component, particularly suited for ISTS like Pendulum. However, its poor performance on other datasets indicates that ISTS methods may not be as effective for general event sequences.
> ```

---

> ### Author Response · Authors · 2024-11-24
> **Continuation of Rebuttal by Authors**
>
> **(Q8) In Figure 3, if no linear trend is observed, what does it suggest?**
> According to Figure 3's caption:
> 'For the Taobao dataset, we do not observe a clear linear trend between \texttt{hpo-val} and the \texttt{test} metric, suggesting the presence of a distribution shift.'
> The absence of correlation between the test metric and \texttt{train-val} on PhysioNet2012 tells us that despite different methods having different validation metrics, they are valued the same on the test set. This suggests that this dataset is unsuitable for model ranking, as it does not effectively differentiate between the performances of different models.
>
> **8(Q9) There is no description of the Taobao dataset used in the assessment.**
>
> We are thankful for this observation and we already added a dataset description.
>
>
> **Overall, we thank the reviewer once again for their valuable feedback and hope that we have successfully addressed all the issues raised. If the reviewer feels that our paper now merits a stronger chance for publication, we would be extremely grateful for a score increase. Your constructive input has been instrumental in improving the quality of our work.**

---

> ### Author Response · Authors · 2024-12-02
> **Some questions and clarifications**
>
> 1. We will include descriptions of datasets with time distributions between events to highlight the irregularity of the data. We should note that this aspect is often not emphasized in papers on Temporal Point Processes, which also focus on discrete, temporally irregular events.
>
> 2. We strongly disagree with the notion that the models used are specific to traditional time series data analysis. For example, most models applied to time series classification are based on convolutional layers and assume that events are uniformly distributed [1].
>
> 3. Furthermore, we invite the reviewer to suggest specialized models for event sequence assessment, and we will gladly include them.
>
> 4. Ultimately, we believe that developing new models for event sequence assessment requires a benchmark like ours.
>
> [1] https://link.springer.com/article/10.1007/s10618-020-00727-3

---

> ### Author Response · Authors · 2024-12-02
>
> >  is there a proof that the selected datasets are significantly more irregular than the ones in current benchmarks?
>
> The essence of time series data lies in the assumption that features are continuous, such as temperature or the coordinates of a pendulum. This is typically true for datasets in weather, medical, and physical domains, as seen in the Pendulum dataset (refer to Appendix Figure 5), which exhibits irregular time stamps and features measurements of continuous coordinates, making it an irregular time series.
>
> However, datasets like MBD, Taobao, Age, and Retail do not contain discrete measurements of continuous processes. For example, categorical features like user actions (e.g., clicks, purchases) in the Taobao dataset cannot be interpolated. Similarly, numerical features like transaction amounts in the MBD dataset are not continuous and cannot be interpolated. The impact on performance has already been discussed: MTAND, designed for Irregularly Sampled Time Series (ISTS), performs well only on the Pendulum dataset, which is an ISTS.

---

### Official Review · Reviewer_aDJA · 2024-11-03

**Soundness:** 3
**Presentation:** 3
**Contribution:** 2
**Rating:** 5
**Confidence:** 3

**Summary:**

This paper proposes a benchmarking set for event sequences, consisting of several known open datasets from various domains. It then provides empirical results with various model architectures (e.g., GRU, Mamba, transformer, etc), on this dataset.

**Strengths:**

Event sequences are a practically relevant problem setting that has indeed gotten relatively less attention from the ML community. Having benchmarking sets is important to progress the field, hence I applaud this effort.

**Weaknesses:**

The authors state on page 3: “One of the primary challenges in benchmarking is ensuring that the datasets used are high quality and **accurately represent the problem domain**. I very much agree with the observation that the value of benchmarking sets lies in their degree of representativity of the problem domain as a whole. Here I wonder if this is the case. Looking at Table 1, I see one dataset with a regression target (Pendulum, which is synthetic), and 5 datasets with a classification target.:
- The first question that can be asked here is whether these should be considered to be the same domain at all. If these are separate problem domains, then perhaps these should instead each have their own benchmark. I would love to see a motivation on why this is a single domain.
- Can we really claim that the domain is covered by this benchmark if there is one regression dataset included? I found that doubtful. Also 5 datasets on the classification-side is still quite limited.
- This work would be much more convincing if the benchmark would have a much larger scale, which would mitigate any concerns about the benchmark sufficiently covering the event sequence domain.

I would like to highlight to the authors the Business Process Intelligence challenge, which has published one event sequence dataset each year since 2011. These are all from the domain of business process management and contain event sequences logged during execution of a business process.

- https://ais.win.tue.nl/bpi/2011/challenge.html until https://ais.win.tue.nl/bpi/2018/challenge.html (change year in url for 2012-2017)

- https://icpmconference.org/2019/icpm-2019/contests-challenges/bpi-challenge-2019/

- https://www.tf-pm.org/competitions-awards/bpi-challenge/2020

There has been extensive literature on machine learning methods on these event sequence datasets (see [2, 3, 4] for surveys). In particular, some of these works already explicitly intend to serve as a benchmark of ML methods on event sequence datasets (e.g., [1] and [3]). I suggest authors incorporate some of these works into their related work. Given that these datasets are all publicly available and exactly fit the event sequence format that is the focus of this paper, it also appears natural to include (some of) these datasets into their benchmark. This may be a step towards addressing the current limitations of insufficient scale.

**References**:

[1] Tax, N., Teinemaa, I., & van Zelst, S. J. (2020). An interdisciplinary comparison of sequence modeling methods for next-element prediction. Software and Systems Modeling, 19(6), 1345-1365.

[3] Teinemaa, I., Dumas, M., Rosa, M. L., & Maggi, F. M. (2019). Outcome-oriented predictive process monitoring: Review and benchmark. ACM Transactions on Knowledge Discovery from Data (TKDD), 13(2), 1-57.

[4] Márquez-Chamorro, A. E., Resinas, M., & Ruiz-Cortés, A. (2017). Predictive monitoring of business processes: a survey. IEEE Transactions on Services Computing, 11(6), 962-977.

[5] Rama-Maneiro, E., Vidal, J. C., & Lama, M. (2021). Deep learning for predictive business process monitoring: Review and benchmark. IEEE Transactions on Services Computing, 16(1), 739-756.

**Questions:**

- Page 5: Is the Holm-Bonferroni correction the right correction? It is not clear why in this setting we would want to control the family-wise error rate rather than the false discovery rate. The latter can be controlled with a Benjamini-Hochberg, and seems to me the more relevant statistical quantity to control. I would like to hear motivation from the authors why instead the decision was made to control the family-wise error rate.

---

> ### Author Response · Authors · 2024-11-24
> **Rebuttal by Authors**
>
> We thank the Reviewer's comment and will incorporate clarifications and modifications upon acceptance.  Below we structure our reply referring to mentioned weaknesses (W) and questions (Q) and we are happy for the following  discussion.
>
>
> **Foreword: First and foremost, we would like to make some clarifications, to support our answers below.**
>
> - While there are numerous other tasks involving event sequences, such as next event and time prediction, interpolation, anomaly detection, or event detection, we focus on assessing a single attribute of an entire sequence.  Our tasks include mainly classification, solely due to the relative ease of finding suitable datasets for this task.
>
> - Moreover, we should emphasise some criteria for our datasets selection. When selecting real-world datasets, we focus on event sequence data with irregular time samples, preferably containing both categorical and numerical features. The most distinct feature of our chosen datasets is the presence of a single target per sequence. This target usually represents a characteristic of the sequence.
>
> - Finally, it is important to note one of our main contributions is a library for model evaluation and a set of benchmarking protocols.  Our code is designed to facilitate the easy addition of new datasets or methods, allowing for their evaluation in a streamlined manner. In our paper we demonstrated how this tool can be used in order to obtain insides about methods and datasets.
>
> **(W1) Domain Similarity of the Pendulum Dataset**
>
> Great question, thank you. While the Pendulum dataset is a regression task without categorical features, making it appear different from other datasets, it falls under our category of datasets by construction, which we discuss in line 801, each sequence is characterised by  target - damping factor.  The primary objective of this dataset is to evaluate the importance of the time component in models (line 126). Although regression tasks can be converted into classification tasks, we believe this would not significantly alter the results. For the lack of regression tasks it is simply due to the fact that there are more dataset for event sequence classification, rather than regression.
>
> **(W2) Domain Coverage of the Benchmark and Business Process Intelligence Challenge**
>
> Thank you for raising this point. We absolutely agree that more datasets would be beneficial. Building a truly outstanding benchmark is not the result of a single paper; it requires sustained effort over time. We hope that some time in the future we can extend our work to the size of UCR Archive [1], which is a result of many years of work. But we need to start somewhere, therefore for now we build a library and a set of benchmarking protocols. Our library allows for the easy addition of new datasets and methods. We hope that these protocols and a library  will allow collaboration with a broad community of researchers in an organised way.
> Moreover, a previously accepted paper at this conference [2] is similar in spirit to our work; the authors proposed a library and seven datasets (one synthetic). In contrast, our work offers more, including an evaluation protocol and a deep analysis of existing methods. We evaluated the effects of different components, scalability, and the time component. Additionally, finding new datasets for our specific needs is more challenging compared to commonly used datasets for TPP. One of the reasons is because we need a target for the entire sequence.
> We appreciate the suggestion to include datasets from the Business Process Intelligence challenge. We evaluated all of the datasets and, unfortunately, they lack a target for the entire sequence. While it is sometimes possible to construct such a target, we were unable to devise one with certainty.
> To address concerns above at the moment we adapted unused data from the MIMIC-III dataset, originally used for classification, to create a new regression target with the same input features but a different target variable. And we are adding one new dataset from [3]. We will update reviews as soon as possible to get the first results. The final results for all the models will be ready for the camera ready version.
> To further highlight challenges in dataset curation. During the rebuttal period  we already evaluated the dataset from [8]. While it fits our domain the dataset is “relatively simple” and we get 0.97 ROC AUC with the RNN model. So, we are not going to include it since it does not help to differentiate models.

---

> ### Author Response · Authors · 2024-11-24
> **Continuation of Rebuttal by Authors**
>
> **(W4) Literature on Event Sequence Datasets**
> Thank you for the suggestion of [4,5,6,7]. As discussed above we face the same challenge with the target. However,  we referenced these papers in the review section to provide a broader look on the domain of event sequence and to distinguish more precise focus of our paper.
>
> **Revised part of the text, which includes references:**
> ```
> Event Sequences is an important domain that encompasses a variety of tasks. There are several distinct research directions that involve Event Sequences. Predictive Process Monitoring (PPM) is a crucial branch of process mining focused on forecasting the future states of ongoing business processes. It involves analyzing event logs to predict various outcomes, such as process completion time, subsequent events, or final outcomes of process instances ~\cite{teinemaa2019outcome, marquez2017predictive, rama2021deep, tax2020interdisciplinary}. One related to PPM but distinct problem is Event detection~\cite{azib2023comprehensive}, accurately identifying specific events is vital for making informed decisions.
> \textbf{Deep Learning-based Temporal Point Processes}, which primarily aims at predicting the type and timing of the next event~\cite{xue2024easytpp}.
> \textbf{Event Sequence Analysis} employs visual methods for data analysis~\cite{zinat2024multi}. This approach focuses on building libraries. In our work, we focus on another important yet understudied task: Event Sequence assessment, which primarily includes the classification or regression of entire sequences. Therefore, our research is distinct from previously studied benchmarks on time series classification and other studies that also address the domain of \ES.
> ```
>
>
>
> **(Q1) Statistical Correction Method**
> Thank you for your valuable remark. We changed the correction method to the Benjamini-Hochberg as you suggested. We will update the main table according to the new procedure. It has affected ranking of some methods on some datasets but it did not change overall results and did not affect previously made statements.
>
> **Overall, we thank the reviewer once again for their valuable feedback and hope that we have successfully addressed all the issues raised. If the reviewer feels that our paper now merits a stronger chance for publication, we would be extremely grateful for a score increase. Your constructive input has been instrumental in improving the quality of our work.**
>
>
> [1] https://link.springer.com/article/10.1007/s10618-020-00727-3
>
> [2] EasyTPP: Towards Open Benchmarking Temporal Point Processes
>
> [3] https://arxiv.org/pdf/2401.03246
>
>
> [4] Tax, N., Teinemaa, I., & van Zelst, S. J. (2020). https://ais.win.tue.nl/bpi/2018/challenge.html. Software and Systems Modeling, 19(6), 1345-1365.
>
> [5] Teinemaa, I., Dumas, M., Rosa, M. L., & Maggi, F. M. (2019). Outcome-oriented predictive process monitoring: Review and benchmark. ACM Transactions on Knowledge Discovery from Data (TKDD), 13(2), 1-57.
>
> [6] Márquez-Chamorro, A. E., Resinas, M., & Ruiz-Cortés, A. (2017). Predictive monitoring of business processes: a survey. IEEE Transactions on Services Computing, 11(6), 962-977.
>
> [7] Rama-Maneiro, E., Vidal, J. C., & Lama, M. (2021). Deep learning for predictive business process monitoring: Review and benchmark. IEEE Transactions on Services Computing, 16(1), 739-756.
>
> [8] https://decan.lexpage.net/files/MSR-2023.pdf

---

> ### Comment · Reviewer_aDJA · 2024-11-26
>
> Given that this paper makes no contributions beyond the benchmark collection of datasets, I remain of the opinion that this work requires extensions to a larger body of datasets to have sufficient meat to substantiate a scientific contribution on its own right. Therefore I maintain my score.
>
> > We appreciate the suggestion to include datasets from the Business Process Intelligence challenge. We evaluated all of the datasets and, unfortunately, they lack a target for the entire sequence.
>
> Note that the literature on outcome-oriented predictive process monitoring, some works of which I have previously listed, study precisely the setting of having a single target for the entire sequence. The datasets from this academic community should be easy to include into EBES, given that their problem setting is the same as in this paper. Benchmark evaluations in this research community typically include datasets from the Business Process Intelligence challenge.
>
> Examples include the BPI'12 and BPI'17 datasets, both of which originate from financial loan application processes from two different financial institutions. Each sequences in these datasets contain a binary label of whether the loan application got rejected or whether it got approved. These are quite natural and well-defined labels.
>
> > To further highlight challenges in dataset curation. During the rebuttal period we already evaluated the dataset from [8]. While it fits our domain the dataset is “relatively simple” and we get 0.97 ROC AUC with the RNN model. So, we are not going to include it since it does not help to differentiate models.
>
> This statement sparks my interest. I wonder why authors believe that a high value on the ROC AUC metric on this dataset in absolute terms would imply that this dataset is not suitable to differentiate models, a task which is about comparison of ROC AUC values in relative terms.
>
> If I would have been convinced that the datasets included in this benchmark truly are the only event sequence datasets available in the public domain, then I may have more accepting of the limited number of datasets that are included in this work. However, with additional availability of the BPI datasets, and the dataset from [8], I find the contribution of this work to be limited and can only recommend authors to extend the number of datasets included in this work.
>
> Having searched a bit more for sequence datasets, I found some more candidates. Including, a dataset of sequences of booked travel destinations at a online travel agency, with one label per sequence being the next booked destination [9].
>
> [9] Dmitri Goldenberg and Pavel Levin. 2021. Booking.com Multi-Destination Trips Dataset. In Proceedings of the 44th International ACM SIGIR Conference on Research and Development in Information Retrieval (SIGIR ’21), July 11–15, 2021, Virtual Event, Canada.

---

> > ### Author Response · Authors · 2024-11-27
> > **Datasets**
> >
> > Thank you for elaborating on the datasets. We will look closer to the previously mentioned and a new addition.
> >
> > In regards with 0.97 AUC, we obtained this metric with a GRU, when no HPO was performed, so our concern is that this dataset will not allow differentiate model performance because most of them will have a high - similar performance. Due to dataset simplicity. So, we will not obtain much information about our methods.

---

> ### Author Response · Authors · 2024-12-02
> **More comments on proposed datasets**
>
> We would like to provide further clarification on the suitability of each dataset:
>
> **Dataset from [8]:**
>
> We would like to note that this dataset contains only 970 sequences in total, which is insufficient for robust evaluation. After 30 runs of hyperparameter optimization, GRU achieved 100% accuracy on the training set, indicating that the dataset is not challenging. The small size also results in poor correlation between subset metrics, making the evaluation too noisy and unreliable.
>
> **Booking:**
>
> We examined the Booking dataset [9] and concluded that it falls outside the scope of our paper.
>
> Firstly, the problem addressed by the authors is the next destination recommendation, which differs significantly from prediction tasks in terms of models, metrics, and approaches. This distinction is evident as the majority of papers utilizing the Booking dataset focus on recommendation systems. The high cardinality of the target space in this dataset is atypical for prediction tasks and necessitates different methodologies [10].
>
> Secondly, even if we disregard the high cardinality of the target space, the problem the authors aim to solve involves predicting the next destination based on a sequence of booked hotels. When a user books the next hotel, the recommendation system updates its predictions and suggests the next city. This scenario is more naturally formalized as a next city prediction problem, which aligns with the studies in Temporal Point Process (TPP) papers and is covered by the EasyTPP benchmark [2].
>
> In conclusion, this dataset is more suitable for recommendation system benchmarks. However, if one overlooks the high cardinality nature of the target space, it could potentially be used for evaluating TPP models.
>
> [10] Petrov, A., & Macdonald, C. (2022, September). Effective and efficient training for sequential recommendation using recency sampling. In Proceedings of the 16th ACM Conference on Recommender Systems (pp. 81-91).

---

### Author Response · Authors · 2024-11-25
**Addressing Reviewer Concerns and Clarifying the Scope**

**We appreciate the valuable feedback provided by all reviewers. In response, we would like to emphasize several points and clarify the scope of our paper, addressing main concerns.**

Our focus is on event sequence assessment, which is  a different domain from multivariate time series and a distinct task from temporal point process modeling, and other tasks applied to event sequences. While assessment (classification or regression) is a generally common task, there is a lack of methodical comparison of these methods in the literature for the event sequence domain. This is despite the high importance of the domain, a point supported by some of the reviewers. Such data is prevalent in real-world applied domains such as finance and medicine; however, no benchmarking work exists which hinders the future research.

**To address this gap, we have taken the following steps:**
1. Developed a library with a unified codebase and interface, allowing seamless incorporation of new datasets and models. The library is open sources. The code is added in supplementary materials to align with conference policies on confidentiality.
2. Outlined a benchmarking protocol to enable a broader audience of researchers to perform method comparisons in a more standardized and structured manner, thereby facilitating research in this direction. Please note that our benchmarking protocol is not a set of strict rules but rather a set of recommendations.
3. Implemented and evaluated a number of methods from the literature in a reproducible manner, providing a fair comparison for our specific problem that did not previously exist in the literature.
4. Carefully curated a set of datasets and conducted the first analysis of a new MDB dataset, which is currently one of the largest of its kind.
5. Our evaluation protocol provides valuable insights into existing datasets and methods.

We wish to highlight the challenge of curating open datasets for our problem, primarily due to the private nature of this data. While there are other open event sequence datasets available, including some proposed by reviewers, we cannot perform event sequence assessment on them because they lack a target for the entire sequence. Furthermore, datasets with classification targets are more common than those with regression targets. For instance, a relevant paper, "EasyTPP Benchmark" [1], accepted to this conference, features only five common datasets for the TPP problem. It is notably easier to find datasets for TPP modeling than for our specific task. Due to the reasons above.

**Finally**, we  aim to continue improving our library and adding new datasets to enhance the scope and applicability of our work. And currently we are working on adding new datasets which will be included into the camera ready version.

[1] https://openreview.net/forum?id=PJwAkg0z7h

---

### Author Response · Authors · 2024-11-29
**Overlooked Contributions and Impact of Our Benchmark**

We appreciate the reviewers' feedback and understand the need to clarify the impact of our work. Our benchmark is more than just a dataset collection; it is a comprehensive framework for rigorous and reproducible research in event sequence assessment.

For each model-dataset combination, we perform approximately 200 runs for hyperparameter optimization and 20 runs for final results, totaling 12,320 training runs across 56 model-dataset pairs. Our library facilitates large-scale evaluations, empowering fair architecture comparison. Our unified interface allows models from different fields to be trained and compared fairly, and our code is highly customizable and reproducible, supporting parallel computing to accelerate the process.

Our work includes thorough ablation studies that provide accurate estimations, setting a high standard for research. Our findings are supported by rigorous experiments, ensuring that our conclusions are robust and reliable.

While the number of datasets is important, our focus is on conducting high-quality research that advances the field. We hope this clarification highlights the broader impact and significance of our work. Thank you for your consideration."

---

### Note · Authors · 2025-02-17

I have read and agree with the venue's withdrawal policy on behalf of myself and my co-authors.

---

### Meta-Review · Area_Chair_7Dm1 · 2024-12-20

**Metareview:**

This paper introduces EBES, a benchmarking tool for classification and regression tasks on event sequences. Event sequences, characterized by irregular time intervals and mixed categorical and numerical features, are prevalent in domains such as healthcare, finance, and user interaction logs. Despite advances in temporal data modeling, the lack of standardized benchmarks has hindered result comparison and potentially misled progress in this field. EBES aims to provide a unified evaluation framework with standardized protocols to address these issues, focusing on sequence-level targets. It includes seven datasets, one of which is a novel synthetic dataset, Pendulum, alongside preprocessed real-world datasets, such as the largest publicly available banking dataset. The tool simplifies the integration of datasets and methods through a unified interface and evaluates model performance using Monte Carlo cross-validation.

The benchmarking results offer an in-depth analysis of datasets, identifying those unsuitable for model comparison and exploring critical aspects such as temporal and sequential modeling, robustness, and scalability. These insights highlight future research directions and aim to standardize reproducible research, expediting advancements in the field.

All the reviewers unanimously commented that the contribution is limited. Specifically, the scope with respect to different models, objectives, and datasets is deemed to be marginal, not sufficient to warrant publication.

**Additional Comments On Reviewer Discussion:**

The authors put significant effort into clarifying the contributions. Nevertheless, the reviewers maintained some of their original criticism about the level of novelty and contribution of the paper.

---

### Decision · Program_Chairs · 2025-01-22

Reject